# Can Variance-Based Regularization Improve Domain Generalization?

## Abstract

Without prior information, domain generalization with only access to multi-domain training data relies on guessing what the test data is. In this work, we consider mild assumptions that there is a distribution over domains and the out-of-distribution data is generated by the shift of the domain distribution. We study a domain-level variance-based regularizer. We show that the variance-regularized method locally approximates the group distributionally robust optimization and embeds the local information into the objective function as a weighting scheme. By taking the empirical domain distribution as an anchor of the location, we propose a weighting correction scheme and provide guarantees of in-distribution generalization. Compared to the Empirical Risk Minimization, we prove the potential benefits of our proposed method but do not observe consistent improvements in general.

## 1 Introduction

Domain generalization [12, 24] is an out-of-distribution (OOD) generalization problem and has drawn much attention recently [33, 38, 30]. Some recent works consider an ambitious goal that generalizes to "absolutely" unseen domain by learning domain-invariant features. From the perspective of theory, the price of such invariant learning methods is the requirement for harsh assumptions or strong prior information, which is necessary to guarantee that the invariance exists and is identifiable. In this work, we assume that there exits a distribution of domains and the OOD test data is generated by the shift of the domain distribution. Then domain generalization is formulated into a distributionally robust optimization problem (DRO, [9, 11, 10]).

Let $\mathbf{z} = (\mathbf{x}, \mathbf{y})$ be a data point consisting of an input vector $\mathbf{x} \in \mathcal{X}$ and the target label $\mathbf{y} \in \mathcal{Y}$. Suppose the training data is structured with respect to a latent domain label:

$$\mathcal{D}_{tr} = \{\mathbf{z}_l, 1 \le l \le m\} = \{\{\mathbf{z}_{i,j}, 1 \le j \le m_i\}, 1 \le i \le n\}, \tag{1}$$

where $m$ is the total sample size , $m_i$ is the sample size of the $i$-th domain and $n$ is the number of domains. We assume that the training domains are randomly drawn from possible domains with a domain distribution $Q$, i.e. $\mathcal{E}_{tr} = \{e_1, e_2, \ldots, e_n\} \subseteq \mathcal{E}$ with $e_i \sim Q$ and the data points under domain $e$ is sampled from the distribution $P_e$. Let $\mathcal{H}$ be the hypothetical space and $h \in \mathcal{H}$ be a model that maps $\mathbf{x} \in \mathcal{X}$ to $h(\mathbf{x}) \in \mathcal{Y}$. The loss function $\ell(\hat{\mathbf{y}}, \mathbf{y}) : \mathcal{Y} \times \mathcal{Y} \to [0, M]$ measures how poorly the output $\hat{\mathbf{y}} = h(\mathbf{x})$ predicts the target $\mathbf{y}$. Denote $\mathcal{F}$ as the collection of the functions $f = \ell(h(\cdot), \cdot) : \mathbf{z} \to [0, M]$ with $h \in \mathcal{H}$. The in-domain expected risk and its sample average approximation ([29]) are denoted by

$$R(f|e_i) = \mathbb{E}_{\mathbf{z} \sim P_{e_i}}[f(\mathbf{z})] \quad \text{and} \quad \hat{R}(f|e_i) = \frac{1}{m_i} \sum_{j=1}^{m_i} f(\mathbf{z}_{i,j}) \tag{2}$$

respectively. The distribution shift between training and test data is characterized by the change of $Q$, while the data distributions $P_e$, $e \in \mathcal{E}$ are fixed.

Submitted to 36th Conference on Neural Information Processing Systems (NeurIPS 2022). Do not distribute.

We study the group distributionally robust optimization problem (group DRO, [18, 26, 28]):

$$\min_{f \in \mathcal{F}} \max_{Q} \mathbb{E}_{\mathbf{z} \sim P}[f(\mathbf{z})], \quad s.t. \quad P = \int P_e Q(\mathrm{d}e), \; D_\phi(Q\|Q_0) \le \rho, \tag{3}$$

where $Q_0$ is a selected domain distribution, $D_\phi(\cdot\|\cdot)$ stands for the $\phi$-divergence ([3, 14]) and the tuning parameter $\rho$ modulates the distribution shift. Throughout this paper, $D_\phi(\cdot\|\cdot)$ is the $\chi^2$-divergence, i.e., $\phi(t) = \frac{1}{2}(t-1)^2$. Sagawa* et al. [28] consider the empirical optimization problem,

$$\min_{f \in \mathcal{F}} \max_{\boldsymbol{q} \in \Delta_n} \sum_{i=1}^n q_i \hat{R}(f|e_i) \quad \text{with} \quad \Delta_n = \left\{ (q_1, \ldots, q_n) : q_i \ge 0, \sum_{i=1}^n q_i = 1 \right\}.$$

Here $\Delta_n$ is the $(n-1)$-dimensional probability simplex. In this case, the parameter $\rho$ is fixed and sufficiently large. For more ambitious goals, Krueger et al. [22] propose the minimax risk extrapolation (MM-REx) that extends the uncertainty region $\Delta_n$ into

$$\tilde{\Delta}_n(\alpha) = \left\{ \boldsymbol{q} = (q_1, \ldots, q_n) : q_i \ge \alpha, \sum_{i=1}^n q_i = 1 \right\},$$

where the parameter $\alpha \in (-\infty, 1/n]$ modulates the uncertainty region. The negative value of $\alpha$ extrapolates risks and encourages robustness to large distribution shifts.

At the sample level, the DRO loss can be asymptotically approximated by the sum of the ERM loss [32] and a variance-based regularizer [16], where the negligible error term converges to zero almost surely. Section 7 in [16] gives general results when the DRO objective is a Hadamard differentiable functional to $P$ and $\mathcal{F}$ is a $P_0$-Donsker class. From the perspective of generalization, the upper bound of the prediction risk may also have a variance-based regularization term that trades between approximation error and estimation error [5, 6, 13, 21]. Sample variance penalization [23] replaces the variance-based regularization with its empirical estimator and gives theoretical guarantees on the prediction performance. To address the computationally intractable problem caused by the non-convexity of the regularizer, Namkoong and Duchi [25] and Duchi and Namkoong [15] investigate the robustly regularized risk, that provides a convex surrogate for variance-regularized loss, and prove finite-sample and asymptotic results characterizing prediction performance. Back to domain generalization problem, Krueger et al. [22] develop a variance-regularized empirical loss (V-REx): $\tilde{R}(f) + \lambda \tilde{V}_{out}(f)$, where

$$\tilde{R}(f) = \frac{1}{n} \sum_{i=1}^n \hat{R}(f|e_i) \quad \text{and} \quad \tilde{V}_{out}(f) = \frac{1}{n} \sum_{i=1}^n \left( \hat{R}(f|e_i) - \tilde{R}(f) \right)^2.$$

Xie et al. [35] prove that with high probability, optimizing the regularized loss $\tilde{R}(f) + \lambda \sqrt{\tilde{V}_{out}(f)}$ is equivalent to solve a MM-REx problem.

In this work, we refine $\tilde{R}(f)$ and $\tilde{V}_{out}(f)$ based on the intuitive understanding of generalization and distribution estimation. Recall the problem in (3). In general, $Q_0$ is the ground-truth domain distribution and $Q$ belongs to a neighborhood of $Q_0$. Therefore, the empirical version of (3) should replace $Q_0$ with its empirical approximation over $\mathcal{E}_{tr}$, i.e.,

$$\hat{\boldsymbol{q}} = (\hat{q}_1, \hat{q}_2, \ldots, \hat{q}_n) = (\frac{m_1}{m}, \ldots, \frac{m_n}{m}).$$

*However, the existing variance-regularized methods directly replace $Q_0$ with a discrete uniform distribution (the center of $\tilde{\Delta}_n(\alpha)$) without considering a consistent and efficient estimator $\hat{\boldsymbol{q}}$.* In the sample variance penalization, this problem does not exist because the discrete uniform distribution on sample points (no tie), i.e. the empirical distribution, is a consistent estimator of the ground-truth data distribution. Consider a new uncertainty region:

$$\mathcal{Q}_{\alpha,\rho}(\hat{\boldsymbol{q}}) = \tilde{\Delta}_n(\alpha) \cap \left\{ \boldsymbol{q} : D_\phi(\boldsymbol{q}\|\hat{\boldsymbol{q}}) \le \rho \right\}.$$

Specifically, any $\boldsymbol{q} = (q_1, \ldots, q_n) \in \mathcal{Q}_{\alpha,\rho}(\hat{\boldsymbol{q}})$ satisfies

$$q_i \ge \alpha, \quad \sum_{i=1}^n q_i = 1, \quad \sum_{i=1}^n \frac{1}{2}(\frac{q_i}{\hat{q}_i} - 1)^2 \hat{q}_i \le \rho.$$

In Section 3.2, we prove that with high probability, the MM-REx problem on $\mathcal{Q}_{\alpha,\rho}(\hat{q})$ can be uniformly equivalent to minimize the variance-regularized empirical loss $\hat{R}(f) + \lambda\sqrt{\hat{V}_{out}(f)}$ where

$$\hat{R}(f) = \sum_{i=1}^{n} \hat{q}_i \hat{R}(f|e_i) \quad \text{and} \quad \hat{V}_{out}(f) = \sum_{i=1}^{n} \hat{q}_i \big(\hat{R}(f|e_i) - \hat{R}(f)\big)^2. \tag{4}$$

Comparing to $\tilde{R}(f)$ and $\tilde{V}_{out}(f)$, the two terms $\hat{R}(f)$ and $\hat{V}_{out}(f)$ just introduce a weighting scheme derived from the empirical domain distribution $\hat{q}$. In addition, the term $\hat{R}(f)$ is the exact ERM loss [32]. In Section 3.1, we investigate the generalization guarantee of the variance-regularized estimator,

$$\hat{f} = \arg\min_{f \in \mathcal{F}} \ \hat{R}(f) + \lambda\sqrt{\hat{V}_{out}(f)}, \tag{5}$$

via the covering number of the function class $\mathcal{F}$. Appendix C also provides a version of the generalization guarantee with localized Rademacher complexities, which may provide tighter generalization bounds in some cases. In Section 4, we consider a general uncertainty region $\mathcal{Q}_{\alpha,\rho}(q_0)$, where the choice of $q_0$ represents a kind of prior knowledge. Similar to the arguments in Section 3, we can also write $q_0$ as a weight assignment and embed it into the variance-regularized loss function. We present a general form of the proposed method and prove that the optimization equivalence in Section 3.2 still holds when we replace $\mathcal{Q}_{\alpha,\rho}(\hat{q})$ with $\mathcal{Q}_{\alpha,\rho}(q_0)$.

Our results clearly show that

- From the perspective of generalization, we propose a weighting correction scheme for variance-regularized domain generalization methods. The proposed method can outperform ERM under some cases, which shows the potential competitive edge of the proposed weighting correction method.

- We do not observe that our method consistently improves ERM under general cases.

- The proposed method is robust to the change of the domain distribution $Q$. From an optimization perspective, it is equivalent to solve a group DRO problem.

## 2   Preliminaries

In this section, we present the rationale for using variance-based regularization to improve the robustness of generalization. Section 2.1 gives two domain adaptation examples that the test data is known. We prove that the standard deviation of risk can bound the generalization gap between training and test data. In Section 2.2, we formulate an invariant learning principle as a hypothesis testing problem. We point out that penalizing the risk variance can protect the null hypothesis: the model is invariant across domains.

### 2.1   Risk variance bounds generalization gap

We present two simple examples to show that penalizing the standard deviation of risk is a natural strategy to improve robustness to the domain distribution shift.

**Risk Interpolation.** In the first example, we assume the test distribution belongs to the convex hull of training domains. This is a typical risk interpolation case. Let $P^*$ be the test distribution. Suppose there exists $q^* = (q_1^*, \cdots, q_n^*) \in \Delta_n$ such that $P^* = \sum_{i=1}^{n} q_i^* P_{e_i}$, where $P_{e_i}, 1 \le i \le n$ are training domains. Then the generalization gap between the training and test data is

$$\text{err}_f = \sum_{i=1}^{n} q_i^* R(f|e_i) - \sum_{i=1}^{n} q_i R(f|e_i) = \sum_{i=1}^{n} (q_i^* - q_i)\Big(R(f|e_i) - \sum_{i=1}^{n} q_i R(f|e_i)\Big),$$

where $q_i = Q(\mathrm{d}e_i)/Q(\mathrm{d}\mathcal{E}_{tr})$ is the proportion of the training domain $e_i$ in the training data. We write $q = (q_1, \ldots, q_n)$. By the Cauchy–Schwarz inequality, we have

$$\text{err}_f \le \sqrt{2D_\phi(q^*\|q)} \times \sqrt{V_{out}(f)}, \tag{6}$$

where $V_{out}(f)$ is the between-domain risk variance over the training domains:

$$V_{out}(f) = \sum_{i=1}^{n} q_i \Big( R(f|e_i) - \sum_{i=1}^{n} q_i R(f|e_i) \Big)^2.$$

Notice that $V_{out}(f)$ only depends the training data. Therefore, it is natural to penalize $\sqrt{V_{out}(f)}$ to obtain a tight upper bound of the test error. The principle here is that if for $\forall e_i \in \mathcal{E}_{tr}$, $R(f|e_i)$ is a constant that only depends on $f$, i.e. $V_{out}(f) = 0$, then changes from $\boldsymbol{q}$ to $\boldsymbol{q}^*$ cannot cause any generalization gap.

**Sub-population Shift.** Recall that the training data in (1) is structured with respect to a latent domain label. In this example, the domain label is the class label $\mathbf{y}$. Therefore, the marginal distribution of $\mathbf{y}$ is different in the training and test data, and the conditional distribution $P(\mathbf{x}|\mathbf{y})$ is the same. Let $\mathcal{Y} = \{1, 2, \ldots, K\}$. Then the generalization gap between the training and test data is

$$
\begin{aligned}
\text{err}_f &= \sum_{k=1}^{K} \mathbb{E}[f(\mathbf{z})|\mathbf{y} = k] \times \big( P_{e'}(\mathbf{y} = k) - P_e(\mathbf{y} = k) \big) \\
&\leq \sqrt{2 D_\phi(P_{e'}(\mathbf{y}) \| P_e(\mathbf{y}))} \times \sqrt{V_{out}(f)},
\end{aligned}
$$

where

$$V_{out}(f) = \sum_{k=1}^{K} P_e(\mathbf{y} = k) \Big( \mathbb{E}[f(\mathbf{z})|\mathbf{y} = k] - \frac{1}{K} \sum_{k=1}^{K} \mathbb{E}[f(\mathbf{z})|\mathbf{y} = k] \Big)^2$$

is the between-class risk variance over the training data. Therefore, the generalization gap is also bounded above by the between-domain risk variance. If the in-class risks are equal, i.e., $\mathbb{E}[f(\mathbf{z})|\mathbf{y} = k] = \mathbb{E}[f(\mathbf{z})|\mathbf{y} = k']$, $\forall k, k' \in \mathcal{Y}$, then the sub-population shift cannot cause generalization gap.

## 2.2 Penalizing risk variance protects invariant models

In this section, we heuristically discuss the relationship between variance-based regularization and invariant learning. The REx principle [22] presents two training goals: **Reducing training risks** and **Increasing the similarity of training risks**. Krueger et al. [22] heuristically explain the utility of V-REx as enforcing the equality of training risks in the limit case $\lambda \to +\infty$. In some experiments, V-REx with small $\lambda$ also shows robust generalization and may outperform ERM. Here we understand this phenomenon by extending the REx principle to the population level:

(i) Minimizing the expected risk $R(f)$;

(ii) Cannot reject the null hypothesis of the test:

$$H_0 : R(f|e) = R(f|e'), \forall e, e' \in \mathcal{E} \quad \text{vs} \quad H_1 : R(f|e) \neq R(f|e'), \exists e, e' \in \mathcal{E}. \quad (7)$$

In general, Principle (i) is achieved by minimizing the ERM loss. Next we show that variance-based regularization is related to the hypothesis testing problem in Principle (ii). Under regular assumptions, one can use the one-way ANOVA F-test to check the hypothesis testing in (7). The F-test statistic is the ratio of the between-domain variance to the in-domain variance, i.e.,

$$F = \frac{\hat{V}_{out}(f)}{\hat{V}(f) - \hat{V}_{out}(f)} \quad \text{with} \quad \hat{V}(f) = \frac{1}{m} \sum_{i=1}^{n} \sum_{j=1}^{m_i} \big( f(\mathbf{z}_{ij}) - \hat{R}(f) \big)^2.$$

Here $\hat{V}_{out}(f)$ and $\hat{R}(f)$ are defined in (4). If $F$ is larger than a threshold, e.g. the $(1 - 5\%)$-quantile of a $F$ distribution, one should reject the null hypothesis. Here $5\%$ is the significance level. If the in-domain variance of a well-trained model is approximately stable, then *penalizing $\hat{V}_{out}(f)$ is equivalent to a constraint that $H_0$ cannot be rejected.* Therefore our proposed method that penalizes $\hat{V}_{out}(f)$ is consistent with the REx principle and the regularization term $\hat{V}_{out}(f)$ is a generalized version of V-REx.

# 3 Variance-Based Regularization

Motivated by Section 2, we study a variance-based regularization method for domain generalization, which minimizes the following empirical loss function:

$$\hat{R}(f) + \lambda\sqrt{\hat{V}_{out}(f)}, \tag{8}$$

where $\lambda$ is a tuning parameter and $\hat{V}_{out}(f)$ is an empirical estimator of the between-domain risk variance. The proposed loss (8) directly optimizes the ERM principle $\hat{R}(f)$, which is different to the recent invariant learning methods that minimize $\tilde{R}(f)$, e.g. Invariant Risk Minimization [1]. The regularization term is slightly different to V-REx: (i) The square-root operator is derived from generalization gap; (ii) *Different to the empirical variance of $R(f|e)$, we penalize the between-domain variance of $f(\mathbf{z})$.*

We consider $Q_0$ in (3) as the training domain distribution and denote the training distribution as $P_0 = \int P_e Q_0(\mathrm{d}e)$. To proceed further, we denote more notations as follows:

$$R(f) = \mathbb{E}_{e \sim Q_0}[R(f|e)] = \mathbb{E}_{\mathbf{z} \sim P_0}[f(\mathbf{z})], \quad V(f) = \mathbb{E}_{\mathbf{z} \sim P_0}\big[(f(\mathbf{z}) - R(f))^2\big],$$

$$V_{in}(f|e) = \mathbb{E}_{\mathbf{z} \sim P_e}\big[(f(\mathbf{z}) - R(f|e))^2\big], \quad V_{out}(f) = \mathbb{E}_{e \sim Q_0}\big[(R(f|e) - R(f))^2\big],$$

where $R(f|e)$ is defined in (2). Here $V_{out}(f)$ is the between-domain variance and $V_{in}(f|e)$ is the in-domain variance of the domain $e \in \mathcal{E}$. According to the decomposition of the total variance, we have

$$V(f) = \mathrm{Var}\big(\mathbb{E}[f(\mathbf{z})|e]\big) + \mathbb{E}\big[\mathrm{Var}(f|e)\big] = V_{out}(f) + \mathbb{E}_{e \sim Q_0}\big[V_{in}(f|e)\big].$$

When $Q_0$ and $P_e$ are replaced by the corresponding empirical distributions, we rewrite $V(f)$, $V_{in}(f|e)$ and $V_{out}(f)$ as $\hat{V}(f)$, $\hat{V}_{in}(f|e)$ and $\hat{V}_{out}(f)$ respectively. In the finite-sample setup, the decomposition of the total variance also holds:

$$\hat{V}(f) = \hat{V}_{out}(f) + \sum_{i=1}^{n} \frac{m_i}{m} \hat{V}_{in}(f|e).$$

## 3.1 Generalization

Since the empirical loss (8) is derived from generalization bounds, we present two versions of the generalization guarantee. The first result depends on the covering number of the function class $\mathcal{F}$. In the appendix, we also derive a version of the generalization bound with localized Rademacher complexities, which can provide more refined uniform generalization bounds in some cases.

We start with the definition of the covering number. Let $\mathcal{F}$ be a collection of bounded functions $f : \mathcal{X} \times \mathcal{Y} \to [0, M]$. Suppose $\mathcal{F}$ is a subset of a metric space with a norm $\|\cdot\|$. We say a collection $\{f^1, \ldots, f^N\} \subseteq \mathcal{F}$ is an $\epsilon$-cover of $\mathcal{F}$ if for each $f \in \mathcal{F}$, there exists $f^i$ such that $\|f - f^i\| \leq \epsilon$. The covering number of $\mathcal{F}$ is

$$N\big(\mathcal{F}, \epsilon, \|\cdot\|\big) \quad := \quad \inf\Big\{N \in \mathbb{N} : \text{there exists a collection } \{f^1, \ldots, f^N\}$$

$$\text{which is an } \epsilon\text{-cover of } \mathcal{F} \text{ with respect to } \|\cdot\|\Big\}.$$

In the following, we use the $\ell^\infty$ norm: $\|f - g\|_\infty = \sup_{z \in \mathcal{X} \times \mathcal{Y}} |f(z) - g(z)|$. Now we are ready to present the following theorem:

**Theorem 1** *Let $n \geq 2$ and $\{\mathbf{z}_{i,j}, 1 \leq i \leq n, 1 \leq j \leq m_i\}$ is an i.i.d sample drawn from $P_0$. Suppose $f(z) \in [0, M]$ for any $f \in \mathcal{F}$ and $z \in \mathcal{X} \times \mathcal{Y}$ and the function class $\mathcal{F}$ has the over number: $N_\epsilon = N(\mathcal{F}, \epsilon, \|\cdot\|_{L^\infty(\mathcal{X} \times \mathcal{Y})})$. Let $0 < \delta < 1$ and*

$$t = \log\frac{(n+2)N_\epsilon}{\delta}, \quad \lambda = \sqrt{\frac{2t}{m-1}}.$$

*Then we have, with probability at least $1 - \delta$,*

$$
\begin{aligned}
R(f) \leq{} & \hat{R}(f) + \lambda\sqrt{\hat{V}_{out}(f)} + \sum_{i=1}^{n} \lambda\sqrt{\frac{(m_i - 1)V_{in}(f|e_i)}{m}} \\
& + \sum_{i=1}^{n} \frac{\sqrt{(m-1)m_i}M\lambda^2}{\sqrt{m(m_i - 1)}} + \frac{(4m - 1)M\lambda^2}{3m} \\
& + \Big(2 + \lambda + \sum_{i=1}^{n} \lambda\sqrt{\frac{(m_i - 1)}{m}}\Big)\epsilon,
\end{aligned}
$$

*holds for every $f \in \mathcal{F}$.*

The proof of Theorem 1 is presented in the Appendix B. In some cases, the covering number-based analysis cannot provide a tight generalization bound [4, 5, 31]. Therefore, we also use the local Rademacher complexity [5] to present the generalization of the proposed variance-based regularization. The details and proof are postponed into Appendix C.

**Why we study In-Distribution generalization?** Theorem 1 provides the generalization guarantee for the in-distribution (ID) generalization rather than the OOD generalization. But its result gives important insights into the OOD generalization. First, the ID error provides a lower bound for the worst-case OOD error since $\mathcal{E}_{tr}$ is a subset of $\mathcal{E}$. Second, some empirical studies of OOD generalization have observed a linear relationship between the ID and OOD test error [27, 17, 19]. Third, some OOD generalization bounds are derived from a domain adaptation framework [8, 7, 2, 36, 37], e.g.,

$$
\text{OOD error} \leq \text{ID error} + \text{error gap} + O(\cdot), \tag{9}
$$

which starts from the ID error and then depicts the error gap. Most recent works focus on minimising the error gap and ignore how their robust (or invariant ) methods increase the ID test error. Fourth, our assumptions are mild and general. We do not impose strong constraint on the test data, e.g. structured generative mechanism, and only assume the domain distribution shift. Therefore, we analyze the ID error of the proposed robust method under mild assumptions.

We denote $f^*$ as the optimal function and let $\hat{f}$ be a solution:

$$
\hat{f} \in \underset{f \in \mathcal{F}}{\arg\min}\ \hat{R}(f) + \lambda\sqrt{\hat{V}_{out}(f)}.
$$

Next we study the excess risk of $\hat{f}$. According to Theorem 1, we obtain the following result.

**Corollary 2** *Suppose the assumptions in Theorem 1 hold. Let $0 < \delta < 1$ and*

$$
t = \log\frac{2N_\epsilon + 2}{\delta}, \quad \lambda = \sqrt{\frac{2t}{m - 1}}.
$$

*Then, with probability at least $1 - \delta$,*

$$
\begin{aligned}
R(\hat{f}) - R(f^*) \leq{} & 2\lambda\sqrt{\frac{(m - 1)V(f^*)}{m}} + \sum_{i=1}^{n} \lambda\sqrt{\frac{m_i\hat{V}_{in}(\hat{f}|e_i)}{m}} \\
& + \Big(2 + \lambda + \sum_{i=1}^{n} \lambda\sqrt{\frac{m_i}{m}}\Big)\epsilon + \lambda^2\frac{4(4m - 1)M}{3m}.
\end{aligned}
$$

**Parametric Example.** Suppose the hypothetical space $\mathcal{F}$ is a class of parametric functions:

$$
\mathcal{F} = \big\{ f_\theta(z) : z \in \mathcal{X} \times \mathcal{Y},\ \theta \in \Theta \subseteq \mathbb{R}^d \big\},
$$

where the parameter set $\Theta$ is bounded. Further, for any data point $\mathbf{z}$, $f_\theta(\mathbf{z})$ is a $L$-Lipschitz function of $\theta$ with respect to $\ell^2$ norm on $\Theta$. Then the covering number is bounded above:

$$
N_\epsilon \leq \Big(1 + \text{diam}(\Theta) \cdot L \cdot \frac{1}{\epsilon}\Big)^d, \quad \text{with} \quad \text{diam}(\Theta) = \sup_{\theta, \theta' \in \Theta} \|\theta - \theta'\|_2.
$$

Then we take

$$\epsilon = \frac{1}{m}, \quad \log N_\epsilon = O(\log m), \quad \lambda = O\left(\sqrt{\frac{\log m}{m}}\right).$$

Therefore, by Corollary 2, with probability at least $1 - \delta$,

$$R(\hat{f}) - R(f^*) \leq 2\lambda\sqrt{\frac{(m-1)V(f^*)}{m}} + \sum_{i=1}^{n}\lambda\sqrt{\frac{m_i\hat{V}_{in}(\hat{f}|e_i)}{m}} + O\left(\frac{\log m}{m}\right). \tag{10}$$

**Potential competitive edge.** The second term on the RHS of (10) contains the empirical in-domain variance $\hat{V}_{in}(\hat{f}|e_i)$. For over-parameterized model, the empirical in-domain variance of $\hat{f}$ can be close to zero. If there exists an optimal function $f^* \in \arg\min_f R(f)$ such that $V(f^*) = 0$, then the term $O(\log m/m)$ dominants the convergence rate of the excess risk.[1] For ERM, the convergence rate of the excess risk is $1/\sqrt{m}$, which is slower than $\log m/m$. *Due to the fast convergence rate, our proposed method can outperform ERM when the sample size $m$ is large enough.*

**Cannot consistently outperform ERM.** If there is no optimal function $f^* \in \arg\min_f R(f)$ satisfies $V(f^*) = 0$, then the first term on the RHS (10) can dominant the excess risk. In this case, the convergence rate of the the excess risk of our method is $\sqrt{\log m/m}$, which is slower than ERM. This implies that *if $V(f^*) > 0$ for $\forall f^* \in \arg\min_f R(f)$, ERM can outperform our method when $m$ is large enough.*

**OOD generalization.** According to Eq. (9), OOD error can be rewritten as a sum of ID error and error gap. The distance between the training and test domain distribution can determine the error gap term under our setup. Furthermore, we only assume that the training and test domain distributions are close but different, and do not impose any structured generative models, such as structural equation models [34] or probabilistic graphical models [20]. In other words, we do not introduce prior information and use mild and general assumptions. Due to the uncertainty of the test data, the error gap should be the worst-case error gap for the domain distribution shift and hypothetical space, which is independent of the estimator. This implies that without prior information, the ID error is a reliable metric to infer the OOD error.

**Non-convexity.** Similar to the Sample Variance Penalization [23], the proposed objective function (8) is in general non-convex and computationally intractable. The proposed regularization term is non-convex even if the loss function is convex. It is still unclear how to actually minimize the variance-regularized objective function. Krueger et al. [22] use a penalty annealing scheme to obtain a good pre-train model. In the Appendix, we empirically show that our method can use random initialization without dropping generalization performance.

## 3.2 Optimization

In this section, we show that minimizing (8) is equivalent to solving a group DRO problem concerning a local neighbourhood of the empirical domain distribution. Let $\boldsymbol{q} = (q_1, q_2, \ldots, q_n)$ be a discrete distributions defined on the domain set $\mathcal{E}_{tr} = \{e_1, e_2, \ldots, e_n\}$. We consider the following optimization problem that minimizes

$$\max_{\boldsymbol{q} \in \mathcal{Q}_{\alpha,\rho}(\hat{\boldsymbol{q}})} \sum_{i=1}^{n} q_i \hat{R}(f|e_i), \tag{11}$$

which is slightly different to group DRO problem because $\mathcal{Q}_\alpha(\hat{\boldsymbol{q}}, \rho)$ is not centered at the uniform discrete distribution. We denote $\lambda = \sqrt{2\rho}$ and rewrite the empirical loss in (8) as

$$\mathcal{L}(f; \rho) = \hat{R}(f) + \sqrt{2\rho\hat{V}_{out}(f)}. \tag{12}$$

The following theorem shows that the objective (11) is bounded by two variance-regularized functions in the form of (12).

---

[1]The factor $\log m$ comes frome the covering number $N_\epsilon$. If the hypothetical space $\mathcal{F}$ only contains finite models, $N_\epsilon$ is a constant and is independent to $m$. Then the convergence rate of the excess risk is $1/m$.

**Theorem 3** *Suppose the training dataset $\mathcal{D}$ and a function $f \in \mathcal{F}$ are given. Let $\rho_+$ be the largest distance between $\hat{q}$ and $q \in \mathcal{Q}_{\alpha,+\infty}(\hat{q})$ and*

$$\rho_- = \frac{\min_i (\alpha/\hat{q}_i - 1)^2 \hat{V}_{out}(f)}{2\big(\min_i \hat{R}(f|e_i) - \hat{R}(f)\big)^2},$$

*then we have*

$$\mathcal{L}(f; \rho_-) \leq \max_{q \in \mathcal{Q}_{\alpha,+\infty}(\hat{q})} \sum_{i=1}^n q_i \hat{R}(f|e_i) \leq \mathcal{L}(f; \rho_+). \tag{13}$$

This second inequality in (13) implies that the optimization problem (11) with $\rho = +\infty$ is always bounded above by the variance-regularized loss with the tuning parameter $\rho_+$. On the other hand, we can also derive a tuning parameter $\rho_-$ depends on the training data and a given model $f$, and then prove that $\mathcal{L}(f; \rho_-)$ is a lower boundary of (11). According to the proof of Theorem 3, one can find that the equality holds:

$$\max_{q \in \mathcal{Q}_{\alpha,\rho}(\hat{q})} \sum_{i=1}^n q_i \hat{R}(f|e_i) = \hat{R}(f) + \sqrt{2\rho \hat{V}_{out}(f)},$$

when the radius $\rho$ satisfies $\rho \leq \rho_-$. If $\hat{V}_{out}(f)$ is nonzero and $\rho$ is given, the equality holds if and only if $\forall e_i \in \mathcal{E}_{tr}$,

$$\alpha \leq \hat{q}_i \Big( \sqrt{\frac{2\rho}{\hat{V}_{out}(f)}} \big( \hat{R}(f|e_i) - \hat{R}(f) \big) + 1 \Big). \tag{14}$$

Therefore, the parameter $\alpha$ and the radius $\rho$ govern each other.

**Sketch of Proof**: We start with a preliminary result: for any $\alpha$ and $\rho$,

$$\max_{q \in \mathcal{Q}_{\alpha,\rho}(\hat{q})} \sum_{i=1}^n q_i \hat{R}(f|e_i) \leq \mathcal{L}(f; \rho),$$

which is directly derived from the Cauchy-Schwarz inequality. By checking the conditions for the equality, we obtain the constraints in (14). Note that $\mathcal{Q}_{\alpha,+\infty}(\hat{q}) \subseteq \mathcal{Q}_{+\infty,\rho_+}(\hat{q})$ since $\rho_+$ is the largest distance between $\hat{q}$ and $q \in \mathcal{Q}_{\alpha,+\infty}(\hat{q})$. Hence the second inequality in (13) is trivial. Let $q_-^*$ be

$$q_-^* = \arg\max_{q \in \mathcal{Q}_{+\infty,\rho_-}(\hat{q})} \sum_{i=1}^n q_i \hat{R}(f|e_i).$$

Furthermore, $q_-^* \in \mathcal{Q}_{\alpha,\rho_-}(\hat{q}) \subseteq \mathcal{Q}_{\alpha,+\infty}(\hat{q})$. Hence the first inequality in (13) holds. $\square$

Theorem 3 shows the equivalence between group DRO and the variance-based regularization in (12). However, the lower bound $\mathcal{L}(f; \rho_-)$ still depends on the training data and model. Next we use concentration inequalities and the covering numbers of $\mathcal{F}$ to derive the uniform results.

**Theorem 4** *Suppose that $\alpha$ is a non-positive scalar and $V'_{out}(f) = V(f) - \sum_i q_i V_{in}(f|e_i) > 0$. For each training domain, both $\hat{q}_i$ and $q_i$ are larger than $\delta > 0$. We write*

$$\rho' = \frac{V'_{out}(f)}{16M^2} \Big( \frac{\alpha}{1 - (n-1)\delta} - 1 \Big)^2.$$

*Let $\tau > 0$ and $0 < \eta < 1$ be two constants. Define*

$$\mathcal{F}_{\tau,\eta} = \{ f \in \mathcal{F} : V(f) \geq \tau, \text{ and } \frac{V_{in}(f|e_i)}{V(f)} \leq \eta, \forall e_i \in \mathcal{E}_{tr} \}.$$

*For any $f \in \mathcal{F}_{\tau,\eta}$, the following expansion uniformly holds:*

$$\max_{q \in \mathcal{Q}_{\alpha,\rho'}(\hat{q})} \sum_{i=1}^n q_i \hat{R}(f|e_i) = \mathcal{L}(f; \rho'),$$

with probability at least $1 - N_{\tau,\eta} \times p$, where $N_{\tau,\eta} = N\left(\mathcal{F}_{\tau,\eta}, \sqrt{\frac{1}{10}(1-\eta)\tau}, \|\cdot\|_{L^\infty(\mathcal{X}\times\mathcal{Y})}\right)$ is the covering number of $\mathcal{F}_{\tau,\eta}$ and

$$
\begin{aligned}
p &= \exp\left(-\frac{m(1-\eta)^2\tau}{32M^2} + \frac{1}{16}\right) + \binom{m+n-1}{n-1}\exp\left(-\frac{m(1-\eta)^2\tau^2}{M^4}\right) \\
&\quad + \sum_{i=1}^{n}\exp\left\{-\frac{1}{2M^2 m_i}\left(\frac{m_i(1-\eta)+4\eta}{1+3\eta}\right)^2\right\}.
\end{aligned}
$$

## 4  General Version

Recall the uncertainty region in (3): $\{Q : D_\phi(Q\|Q_0) \leq \rho\}$. In Section 3, $Q_0$ is the ground-truth domain distribution. In fact, it can be a selected anchor distribution closed to the target test domain. The choice of $Q_0$ can be regarded as a kind of prior knowledge and the hyperparameter $\rho$ represents how strong is the confidence in the prior. We formulate the finite-sample optimization problem as

$$
\max_{\boldsymbol{q}\in\mathcal{Q}_{\alpha,\rho}(\boldsymbol{q}_0)} \sum_{i=1}^{n} q_i \hat{R}(f|e_i), \tag{15}
$$

where $\boldsymbol{q}_0$ is the conditional distribution of $e$ given $e \in \mathcal{E}_{tr}$, which is derived from $Q_0$. In this problem, the uncertainty region $\mathcal{Q}_{\alpha,\rho}(\boldsymbol{q}_0)$ is centered at a discrete distribution $\boldsymbol{q}_0$ rather than the uniform distribution or the empirical distribution $\hat{\boldsymbol{q}}$. Therefore, we can manually select $\boldsymbol{q}_0$ to introduce the prior information.

According to the proof of Theorem 3 in the Appendix, the optimization equivalence in Section 3.2 also holds when we replace $\mathcal{Q}_{\alpha,\rho}(\hat{\boldsymbol{q}})$ with $\mathcal{Q}_{\alpha,\rho}(\boldsymbol{q}_0)$. To proceed further, we rewrite $\mathcal{L}(f;\rho,\boldsymbol{q}_0) = \hat{R}(f,\boldsymbol{q}_0) + \sqrt{2\rho\hat{V}_{out}(f,\boldsymbol{q}_0)}$ with

$$
\hat{R}(f,\boldsymbol{q}_0) = \sum_{i=1}^{n} q_{0,i}\hat{R}(f|e_i) \quad\text{and}\quad \hat{V}_{out}(f,\boldsymbol{q}_0) = \sum_{i=1}^{n} q_{0,i}\left(\hat{R}(f|e_i) - \hat{R}(f,\boldsymbol{q}_0)\right)^2.
$$

Then we restate Theorem 3 as the following general version.

**Theorem 5** *Given the training dataset and a function $f \in \mathcal{F}$, then for any distribution $\boldsymbol{q}_0$, the inequality always holds:*

$$
\max_{\boldsymbol{q}\in\mathcal{Q}_{\alpha,\rho}(\boldsymbol{q}_0)} \sum_{i=1}^{n} q_i \hat{R}(f|e_i) \leq \mathcal{L}(f;\rho,\boldsymbol{q}_0).
$$

*If the between-domain variance $\hat{V}_{out}(f,\boldsymbol{q}_0)$ is non-zero, the equality holds if and only if $\forall e_i \in \mathcal{E}_{tr}$,*

$$
\alpha \leq q_{0,i}\left(\sqrt{\frac{2\rho}{\hat{V}_{out}(f,\boldsymbol{q}_0)}}\left(\hat{R}(f|e_i) - \hat{R}(f,\boldsymbol{q}_0)\right) + 1\right).
$$

*On the other hand, if $\alpha$ is fixed, the equality holds when the radius of $\mathcal{Q}_{\alpha,\rho}(\boldsymbol{q}_0)$ satisfies*

$$
\rho \leq \frac{\min_i(\alpha/q_{0,i}-1)^2\hat{V}_{out}(f,\boldsymbol{q}_0)}{2\left(\min_i \hat{R}(f|e_i) - \hat{R}(f,\boldsymbol{q}_0)\right)^2}.
$$

This result shows the equivalence between the optimization problem (15) and the variance-regularized loss $\mathcal{L}(f;\rho,\boldsymbol{q}_0)$. Therefore, for unbalanced domains and any given prior $\boldsymbol{q}_0$, we can still use the variance-based regularization to approximate the DRO problem. Please refer to the Appendix for the complete proof of Theorem 5.

## 5  Conclusion

In this work, we study a variance-based regularization method for domain generalization. We prove the guarantees for in-distribution generalization and figure out the potential benefits of our proposed method compared to ERM. Our proposed objective function is non-convex and the optimization procedure is computationally intractable. The learnt model can be highly dependent on initialization or pretraining. In future work, we will consider combining generalization bounds with specific optimization algorithms to seek fine-grained generalization guarantees.

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
