# A  Experiments

In this section, we present empirical evidence to verify our theoretical results that under mild and general assumptions, *the proposed weighting correction scheme in (5) has better ID generalization guarantees than the existing variance-regularized domain generalization methods.* By Theorem 4, we reformulate the objective function in (5) into a batch version. Then we consider the following two settings:

- **Balanced batch.** This is the standard operation in DomainBed [18] and is commonly used by the existing variance-based regularization methods. In each iteration, the same number of data points are randomly drawn from each training domain to form a batch.

- **Unbalanced batch.** (Our method) In this setting, we randomly draw data points from each training domain with equal proportions, such that the proportion of each domain in one batch is the same as the proportion of the domains in the entire training data.

We consider three variance-regularized domain generalization methods.

- **Variance.** The V-REx regularization [25] penalizes the domain-level variance of risk without considering the empirical domain distribution. The V-REx estimator is

$$\hat{f} = \underset{f \in \mathcal{F}}{\arg\min}\, \tilde{R}(f) + \lambda \tilde{V}_{out}(f),$$

where

$$\tilde{R}(f) = \frac{1}{n}\sum_{i=1}^{n}\hat{R}(f|e_i) \quad \text{and} \quad \tilde{V}_{out}(f) = \frac{1}{n}\sum_{i=1}^{n}\left(\hat{R}(f|e_i) - \tilde{R}(f)\right)^2.$$

- **Standard Deviation.** We also consider the RVP Regularization [41], which slightly changes the penalty term of V-REx into the domain-level standard deviation of risk. Xie et al. [41] provides an understanding of generalization from the perspective of quantile regression and shows that RVP locally approximates DRO. We *remove the scheme of penalty annealing* since it is not involved in group DRO. The RVP estimator is

$$\hat{f} = \underset{f \in \mathcal{F}}{\arg\min}\, \tilde{R}(f) + \lambda\sqrt{\tilde{V}_{out}(f)}.$$

- **Weighting Correction.** Our proposed method introduces the empirical domain distribution as a weighting scheme into the objective function. We also *remove the scheme of penalty annealing*. Our proposed estimator is

$$\hat{f} = \underset{f \in \mathcal{F}}{\arg\min}\, \hat{R}(f) + \lambda\sqrt{\hat{V}_{out}(f)},$$

where

$$\hat{R}(f) = \sum_{i=1}^{n}\hat{q}_i\hat{R}(f|e_i) \quad \text{and} \quad \hat{V}_{out}(f) = \sum_{i=1}^{n}\hat{q}_i\big(\hat{R}(f|e_i) - \hat{R}(f)\big)^2.$$

417 **Implementation Details.** We consider two datasets: PACS [26] and VLCS [37] We use ResNet50
418 as neural network architecture [19] and start from a pretrained model on ImageNet [32]. In order
419 to fairly evaluate the different regularization, we follow the DomainBed Benchmark to randomly
420 select 20 groups of hyperparameter combinations and repeated the experiment three times for each
421 hyperparameter group. The model is selected according to the training domain validation accuracy,
422 that is the in-distribution validation accuracy. The hyper-parameters includes batch size, learning
423 rate, weight decay, iterations of penalty annealing, and the regularization parameter $\lambda$. The other
424 experimental settings are the same as those in Gulrajani and Lopez-Paz [18].

425 The results are reported in Table 1. One can find that the improvement of the weighting correction
426 scheme is statistically significant compared to the original VREx method.

Table 1: The ID prediction accuracy on PACS and VLCS.

| PACS | | | | | | |
|---|---|---|---|---|---|---|
| **Balance** | **Regularization Method** | **A** | **C** | **P** | **S** | **Avg** |
| $\times$ | Weighting Correction | $96.9 \pm 0.1$ | $\mathbf{97.0 \pm 0.2}$ | $\mathbf{96.4 \pm 0.1}$ | $\mathbf{97.2 \pm 0.2}$ | $\mathbf{\color{red}96.9 \pm 0.1}$ |
| $\times$ | Standard Deviation | $67.4 \pm 0.3$ | $54.9 \pm 1.0$ | $51.0 \pm 1.4$ | $82.4 \pm 0.4$ | $63.9 \pm 0.2$ |
| $\times$ | Variance | $20.5 \pm 0.5$ | $19.6 \pm 0.2$ | $20.5 \pm 0.5$ | $36.7 \pm 5.9$ | $24.3 \pm 1.7$ |
| $\checkmark$ | Standard Deviation | $82.0 \pm 0.6$ | $81.3 \pm 0.1$ | $77.2 \pm 0.6$ | $86.1 \pm 0.5$ | $81.7 \pm 0.4$ |
| $\checkmark$ | Variance | $\mathbf{97.0 \pm 0.1}$ | $96.6 \pm 0.1$ | $96.2 \pm 0.2$ | $97.0 \pm 0.2$ | $\color{red}96.7 \pm 0.1$ |
| **VLCS** | | | | | | |
| **Balance** | **Regularization Method** | **C** | **L** | **S** | **V** | **Avg** |
| $\times$ | Weighting Correction | $\mathbf{81.9 \pm 0.2}$ | $\mathbf{87.6 \pm 0.1}$ | $\mathbf{85.7 \pm 0.3}$ | $\mathbf{83.4 \pm 0.0}$ | $\mathbf{\color{red}84.7 \pm 0.0}$ |
| $\times$ | Standard Deviation | $43.7 \pm 0.0$ | $45.0 \pm 0.2$ | $48.2 \pm 0.7$ | $46.1 \pm 0.5$ | $45.8 \pm 0.3$ |
| $\times$ | Variance | $53.8 \pm 1.5$ | $49.1 \pm 0.9$ | $53.3 \pm 0.9$ | $52.7 \pm 1.7$ | $52.2 \pm 0.5$ |
| $\checkmark$ | Standard Deviation | $48.0 \pm 2.3$ | $47.5 \pm 0.7$ | $48.9 \pm 0.5$ | $53.6 \pm 0.4$ | $49.5 \pm 0.7$ |
| $\checkmark$ | Variance | $81.3 \pm 0.2$ | $87.3 \pm 0.2$ | $85.4 \pm 0.5$ | $83.1 \pm 0.2$ | $\color{red}84.3 \pm 0.2$ |

## B    Proof of Theorem 1

428 **Theorem.** *Let $n \geq 2$ and $\{\mathbf{z}_{i,j}, 1 \leq i \leq n, 1 \leq j \leq m_i\}$ is an i.i.d sample drawn from $P_0$.. Suppose*
429 *the function set $\mathcal{F}$ has cover numbers*

$$N_\epsilon = N(\mathcal{F}, \epsilon, \|\cdot\|_{L^\infty(\mathcal{X} \times \mathcal{Y})}),$$

430 *and for any $f \in \mathcal{F}$ and $z \in \mathcal{X} \times \mathcal{Y}$, $f(z) \in [0, M]$. Then we have for every $f \in \mathcal{F}$,*

$$
\begin{aligned}
R(f) \quad \leq \quad & \hat{R}(f) + \sqrt{\frac{2\hat{V}_{out}(f)t}{m-1}} + \sum_{i=1}^{n} \sqrt{\frac{2(m_i-1)V_{in}(f|e_i)t}{m(m-1)}} \\
& + \sum_{i=1}^{n} \frac{2\sqrt{m_i}Mt}{\sqrt{m(m_i-1)(m-1)}} + \frac{2(4m-1)Mt}{3m(m-1)} \\
& + \Big(2 + \sqrt{\frac{2t}{m-1}} + \sum_{i=1}^{n} \sqrt{\frac{2(m_i-1)t}{m(m-1)}}\Big)\epsilon,
\end{aligned}
$$

431 *with probability at least $1 - (n+2)N_\epsilon \exp(-t)$.*

432 **Proof:** By the Bernstein inequality, with probability at least $1 - \exp(-t)$,

$$R(f) \quad \leq \quad \hat{R}(f) + \sqrt{\frac{2V(f)t}{m}} + \frac{2Mt}{3m},$$

433 holds for any given function $f$. Furthermore, by Theorem 10 in [27],

$$\sqrt{V(f)} \quad \leq \quad \sqrt{\frac{m}{m-1}\hat{V}(f)} + \frac{\sqrt{2mM^2t}}{m-1},$$

434    holds with probability larger than $1 - \exp(-t)$. Then,

$$
\begin{aligned}
R(f) &\leq \hat{R}(f) + \sqrt{\frac{2\hat{V}(f)t}{m-1}} + \frac{2Mt}{m-1} + \frac{2Mt}{3m}, \\
&= \hat{R}(f) + \sqrt{\frac{2\hat{V}(f)t}{m-1}} + \frac{2(4m-1)Mt}{3m(m-1)},
\end{aligned}
$$

435    holds with probability larger than $1 - 2\exp(-t)$. According to the decomposition of the total variance,
436    we know that

$$
\begin{aligned}
\hat{V}(f) &= \frac{1}{m}\sum_{i=1}^{n}\sum_{j=1}^{m_i}\left(f(\mathbf{z}_{i,j}) - \hat{R}(f|e_i) + \hat{R}(f|e_i) - \hat{R}(f)\right)^2 \\
&= \frac{1}{m}\sum_{i=1}^{n}\sum_{j=1}^{m_i}\left(f(\mathbf{z}_{i,j}) - \hat{R}(f|e_i)\right)^2 + \left(\hat{R}(f|e_i) - \hat{R}(f)\right)^2 \\
&= \sum_{i=1}^{n}\frac{m_i}{m}\hat{V}_{in}(f|e_i) + \sum_{i=1}^{n}\hat{q}_i\left(\hat{R}(f|e_i) - \hat{R}(f)\right)^2 \\
&= \sum_{i=1}^{n}\frac{m_i}{m}\hat{V}_{in}(f|e_i) + \hat{V}_{out}(f).
\end{aligned}
$$

437    Hence, with probability larger than $1 - 2\exp(-t)$,

$$
R(f) \leq \hat{R}(f) + \sqrt{\frac{2\hat{V}_{out}(f)t}{m-1}} + \sum_{i=1}^{n}\sqrt{\frac{2m_i\hat{V}_{in}(f|e_i)t}{m(m-1)}} + \frac{2(4m-1)Mt}{3m(m-1)}.
$$

438    By applying Theorem 10 in [27],

$$
\sqrt{V_{in}(f|e_i)} \geq \sqrt{\frac{m_i}{m_i - 1}\hat{V}_{in}(f|e_i)} - \frac{\sqrt{2m_i M^2 t}}{m_i - 1},
$$

439    holds with probability smaller than $\exp(-t)$. Hence we have,

$$
\begin{aligned}
R(f) \leq \hat{R}(f) &+ \sqrt{\frac{2\hat{V}_{out}(f)t}{m-1}} + \sum_{i=1}^{n}\sqrt{\frac{2(m_i - 1)V_{in}(f|e_i)t}{m(m-1)}} \\
&+ \sum_{i=1}^{n}\frac{2\sqrt{m_i}Mt}{\sqrt{m(m_i-1)(m-1)}} + \frac{2(4m-1)Mt}{3m(m-1)},
\end{aligned}
$$

440    holds with probability larger than $1 - (2 + n)\exp(-t)$.

441    Next, we consider a set of functions $\{f^1, \ldots, f^{N_\epsilon}\}$, which is a minimal $\epsilon$-cover of the function space
442    $\mathcal{F}$ of size

$$
N_\epsilon = N(\mathcal{F}, \epsilon, \|\cdot\|_{L^\infty(\mathcal{X}\times\mathcal{Y})}).
$$

443    Then, for any $f \in \mathcal{F}$, there exists $f^j$, $1 \leq j \leq N_\epsilon$ such that $\|f - f^j\|_{L^\infty(\mathcal{X}\times\mathcal{Y})} \leq \epsilon$. Therefore,

$$
\begin{aligned}
R(f) \leq\ & R(f^j) + \epsilon \\
\leq\ & \hat{R}(f^j) + \sqrt{\frac{2\hat{V}_{out}(f^j)t}{m-1}} + \sum_{i=1}^{n}\sqrt{\frac{2(m_i - 1)V_{in}(f^j|e_i)t}{m(m-1)}} \\
& + \sum_{i=1}^{n}\frac{2\sqrt{m_i}Mt}{\sqrt{m(m_i-1)(m-1)}} + \frac{2(4m-1)Mt}{3m(m-1)} + \epsilon,
\end{aligned}
$$

444    with probability larger than $1 - (2 + n)\exp(-t)$. Notice that $\hat{R}(f^j) \leq \hat{R}(f) + \epsilon$ and

$$
\begin{aligned}
\sqrt{\hat{V}_{out}(f^j)} &\leq \sqrt{\hat{V}_{out}(f)} + \sqrt{\hat{V}_{out}(f^j - f)} \leq \sqrt{\hat{V}_{out}(f)} + \epsilon, \\
\sqrt{\hat{V}_{in}(f^j|e_i)} &\leq \sqrt{\hat{V}_{in}(f|e_i)} + \sqrt{\hat{V}_{in}(f^j - f|e_i)} \leq \sqrt{\hat{V}_{in}(f|e_i)} + \epsilon.
\end{aligned}
$$

Therefore, for every $f \in \mathcal{F}$,

$$
\begin{aligned}
R(f) \quad \leq \quad & \hat{R}(f) + \sqrt{\frac{2\hat{V}_{out}(f)t}{m-1}} + \sum_{i=1}^{n}\sqrt{\frac{2(m_i-1)V_{in}(f|e_i)t}{m(m-1)}} \\
& + \sum_{i=1}^{n}\frac{2\sqrt{m_i}Mt}{\sqrt{m(m_i-1)(m-1)}} + \frac{2(4m-1)Mt}{3m(m-1)} \\
& + \Big(2 + \sqrt{\frac{2t}{m-1}} + \sum_{i=1}^{n}\sqrt{\frac{2(m_i-1)t}{m(m-1)}}\Big)\epsilon,
\end{aligned}
$$

holds with probability at least $1 - (n+2)N_\epsilon \exp(-t)$.

$\square$

Let

$$
t = \log\frac{(n+2)N_\epsilon}{\delta}, \quad \text{and} \quad \lambda = \sqrt{\frac{2t}{m-1}}.
$$

Then we have, with probability at least $1 - \delta$,

$$
\begin{aligned}
R(f) \quad \leq \quad & \hat{R}(f) + \lambda\sqrt{\hat{V}_{out}(f)} + \sum_{i=1}^{n}\lambda\sqrt{\frac{(m_i-1)V_{in}(f|e_i)}{m}} \\
& + \sum_{i=1}^{n}\frac{\sqrt{(m-1)m_i}M\lambda^2}{\sqrt{m(m_i-1)}} + \frac{(4m-1)M\lambda^2}{3m} \\
& + \Big(2 + \sqrt{\frac{2t}{m-1}} + \sum_{i=1}^{n}\sqrt{\frac{2(m_i-1)t}{m(m-1)}}\Big)\epsilon,
\end{aligned}
$$

for every $f \in \mathcal{F}$. Hence Theorem 1 is proved.

$\square$

# C   More results for Theorem 1

In this section, we use the local Rademacher complexity [5] to present the generalization of the proposed variance-based regularization. We start with the definition of the sub-root function, which can be used to bound the local Rademacher complexity.

**Definition 6 ([5], Definition 3.1)** *A function $\psi : [0,\infty) \to [0,\infty)$ is sub-root if it is non-negative, non-decreasing and if $r \mapsto \psi(r)/\sqrt{r}$ is non-increasing for $r > 0$.*

For any nontrivial sub-root function $\psi$, i.e., not the constant function $\psi \equiv 0$, it is continuous and has a unique positive fix point $r^* = \psi(r^*)$. In addition, for all $r > 0$, $r \geq \psi(r)$ if and only if $r^* \leq r$. We consider the local Rademacher complexity:

$$
\mathbb{E}\big[\mathcal{R}_n(\{cf(z) : f \in \mathcal{F}, c \in [0,1], \text{ and } \mathbb{E}_{\mathbf{z}\sim P}[c^2 f^2(\mathbf{z})] \leq r\})\big] \tag{16}
$$

which is also used by [16]. Here the notation $\mathbb{E}$ in (16) takes the expectations with respect to the Rademacher random variables. We denote a sub-root function $\psi_m(r)$ as an upper bound of the localized Rademacher complexity:

$$
\psi_m(r) \quad \geq \quad \mathbb{E}\big[\mathcal{R}_n(\{cf(z) : f \in \mathcal{F}, c \in [0,1], \text{ and } \mathbb{E}_{\mathbf{z}\sim P}[c^2 f^2(\mathbf{z})] \leq r\})\big]. \tag{17}
$$

The solution of $\psi_m(r) = r$ is denoted as $r_m^*$. When the distribution $P$ is replaced by $P_{e_i}$, the upper bound sub-root function and the corresponding fixed point are written as $\psi_{m,i}$ and $r_{m,i}^*$.

**Theorem 7** *Let $\mathcal{F}$ be a collection of bounded functions $f : \mathcal{X} \times \mathcal{Y} \to [0,M]$ satisfying the localization inequality (17) for some sub-root function $\psi_m(r)$ $(\psi_{m,i}(r))$ with root $r_m^*$ $(r_{m,i}^*)$. Let*

$$
B_m = \frac{1}{m}(t + \log\lceil\log\frac{m}{t}\rceil) \quad \text{and} \quad C_m = 2((2e + 84M)B_m + 36r_m^*),
$$

where $\lceil \cdot \rceil$ stands for the ceiling function. Then, for every $f \in \mathcal{F}$,

$$
\begin{aligned}
R(f) \quad \leq \quad & (1 + \sqrt{2C_m})\left( \hat{R}(f) + \frac{\sqrt{2C_m}}{1 + \sqrt{2C_m}} \sqrt{\hat{V}_{out}(f)} \right) \\
& + \sqrt{\frac{3}{2} C_m \sum_{i=1}^{n} \frac{m_i}{m} \mathbb{E}[f^2(\mathbf{z})|e_i]} \\
& + \sqrt{144 C_m M^2 \sum_{i=1}^{n} \frac{m_i}{m} r_{m,i}^* + C_m \frac{nMt}{m}\left(4 + \frac{7}{3}M\right)} \\
& + \sqrt{144 C_m M^2 r_m^* + \frac{C_m Mt}{m}\left(4 + \frac{7}{3}M\right) + 6r_m^* + 14MB_m},
\end{aligned}
$$

with probability at least $1 - (2+n)\exp(-t)$.

**Proof:** Before proving the theorem, we state a useful lemma that provides a version of uniform Bernstein's inequality by measuring the complexity of the localized functions that near the optimum of an empirical risk.

**Lemma 8** [[16], Lemma 17 and Lemma 18] *Let $\mathcal{F}$ be a collection of bounded functions $f : \mathcal{X} \times \mathcal{Y} \to [0, M]$ satisfying the localization inequality (17) for some sub-root function $\psi_m(r)$ with root $r_m^*$. Let*

$$
B_m = \frac{1}{m}(t + \log\lceil \log \frac{m}{t} \rceil) \quad \text{and} \quad \eta > 0.
$$

*Then with probability at least $1 - \exp(-t)$, for every $f \in \mathcal{F}$,*

$$
R(f) - \hat{R}(f)| \leq \left(\sqrt{2eB_m} + 6\sqrt{r_m^* + \frac{7}{3}MB_m}\right)\sqrt{\mathbb{E}[f^2]} + 6r_m^* + 14MB_m, \tag{18}
$$

$$
\mathbb{E}[f^2] \leq \hat{\mathbb{E}}[f^2] + \frac{1}{\eta}\hat{\mathbb{E}}[f^2] + 72M^2(1+\eta)r_m^* + \frac{Mt}{m}\left(4 + \frac{7}{3}M\right), \tag{19}
$$

*and*

$$
\hat{\mathbb{E}}[f^2] \leq \mathbb{E}[f^2] + \frac{\eta}{\eta+1}\mathbb{E}[f^2] + 72M^2(1+\eta)r_m^* + \frac{Mt}{m}\left(4 + \frac{7}{3}M\right), \tag{20}
$$

*where $\hat{\mathbb{E}}[f^2(\mathbf{z})] = \frac{1}{m}\sum_{i=1}^{n}\sum_{j=1}^{m_i} f^2(\mathbf{z}_{i,j})$.*

**Proof:** Please refer to Appendix D.1 and D.2 of [16] for a complete proof of the lemma.

$\square$

Now we turn to the proof of Theorem 7. We denote $C_m = 2((2e + 84M)B_m + 36r_m^*)$. It is easy to see that

$$
\begin{aligned}
& (\sqrt{2eB_m} + 6\sqrt{r_m^* + 7MB_m/3})^2 \\
= \quad & 2eB_m + (36r_m^* + 84MB_m) + 2\sqrt{2eB_m}\sqrt{36r_m^* + 84MB_m} \\
\leq \quad & 2(2eB_m + 36r_m^* + 84MB_m) = 2C_m.
\end{aligned}
$$

Then the inequality (18) implies that for every $f \in \mathcal{F}$,

$$
R(f) \quad \leq \quad \hat{R}(f) + \sqrt{C_m \mathbb{E}[f^2(\mathbf{z})]} + 6r_m^* + 14MB_m,
$$

holds with probability at least $1 - \exp(-t)$. By (19) with $\eta = 1$, we have for all $f \in \mathcal{F}$,

$$
R(f) \leq \hat{R}(f) + \sqrt{2C_m \hat{\mathbb{E}}[f^2(\mathbf{z})]} + \sqrt{144 C_m M^2 r_m^* + \frac{C_m Mt}{m}\left(4 + \frac{7}{3}M\right)} + 6r_m^* + 14MB_m,
$$

with probability at least $1 - 2\exp(-t)$. Notice that

$$
\begin{aligned}
\hat{\mathbb{E}}[f^2(\mathbf{z})] &= \frac{1}{m}\sum_{i=1}^{n}\sum_{j=1}^{m_i}\left(f(\mathbf{z}_{i,j}) - \hat{R}(f|e_i) + \hat{R}(f|e_i) - \hat{R}(f) + \hat{R}(f)\right)^2 \\
&= \frac{1}{m}\sum_{i=1}^{n}\sum_{j=1}^{m_i}\left(f(\mathbf{z}_{i,j}) - \hat{R}(f|e_i)\right)^2 + \left(\hat{R}(f|e_i) - \hat{R}(f)\right)^2 + \hat{R}(f)^2 \\
&= \hat{R}(f)^2 + \hat{V}_{out}(f) + \sum_{i=1}^{n}\frac{m_i}{m}\hat{V}_{in}(f|e_i).
\end{aligned}
$$

Then we have

$$
\begin{aligned}
R(f) \le\ & (1+\sqrt{2C_m})\hat{R}(f) + \sqrt{2C_m\hat{V}_{out}(f)} + \sqrt{C_m\sum_{i=1}^{n}\frac{m_i}{m}\hat{V}_{in}(f|e_i)} \\
& + \sqrt{144C_mM^2r_m^* + \frac{C_mMt}{m}\left(4 + \frac{7}{3}M\right)} + 6r_m^* + 14MB_m.
\end{aligned}
$$

Next we deal with the upper bound of $\hat{V}_{in}(f|e_i)$. We denote

$$
\hat{\mathbb{E}}[f^2|e_i] = \frac{1}{m_i}\sum_{j=1}^{m_i}f^2(\mathbf{z}_{i,j}).
$$

According to (20), for every $f \in \mathcal{F}$,

$$
\hat{\mathbb{E}}[f^2(\mathbf{z})|e_i] \le \frac{2\eta+1}{\eta+1}\mathbb{E}[f^2(\mathbf{z})|e_i] + 72M^2(1+\eta)r_{m,i}^* + \frac{Mt}{m_i}\left(4 + \frac{7}{3}M\right),
$$

holds with probability at least $1 - \exp(-t)$. Let $\eta = 1$. Then, for every $f \in \mathcal{F}$,

$$
\begin{aligned}
R(f) \le\ & (1+\sqrt{2C_m})\hat{R}(f) + \sqrt{2C_m\hat{V}_{out}(f)} + \sqrt{\frac{3}{2}C_m\sum_{i=1}^{n}\frac{m_i}{m}\mathbb{E}[f^2(\mathbf{z})|e_i]} \\
& + \sqrt{144C_mM^2\sum_{i=1}^{n}\frac{m_i}{m}r_{m,i}^* + C_m\frac{nMt}{m}\left(4 + \frac{7}{3}M\right)} \\
& + \sqrt{144C_mM^2r_m^* + \frac{C_mMt}{m}\left(4 + \frac{7}{3}M\right)} + 6r_m^* + 14MB_m,
\end{aligned}
$$

with probability at least $1 - (2+n)\exp(-t)$.

$\square$

# D   Proof of Corollary 2

**Corollary.** *Let $n \ge 2$ and $\{\mathbf{z}_{i,j}, 1 \le i \le n, 1 \le j \le m_i\}$ is an i.i.d sample. Suppose the function set $\mathcal{F}$ has cover numbers*

$$
N_\epsilon = N(\mathcal{F}, \epsilon, \|\cdot\|_{L^\infty(\mathcal{X}\times\mathcal{Y})}),
$$

*and for any $f \in \mathcal{F}$ and $z \in \mathcal{X} \times \mathcal{Y}$, $f(z) \in [0, M]$. Let*

$$
t = \log\frac{2N_\epsilon + 2}{\delta}, \quad and \quad \lambda = \sqrt{\frac{2t}{m-1}}.
$$

*Then, with probability at least $1 - \delta$,*

$$
\begin{aligned}
R(\hat{f}) - R(f^*) \le\ & 2\lambda\sqrt{\frac{(m-1)V(f^*)}{m}} + \sum_{i=1}^{n}\lambda\sqrt{\frac{m_i\hat{V}_{in}(\hat{f}|e_i)}{m}} \\
& + \left(2 + \lambda + \sum_{i=1}^{n}\lambda\sqrt{\frac{m_i}{m}}\right)\epsilon + \lambda^2\frac{4(4m-1)M}{3m}.
\end{aligned}
$$

**Proof:** According to the proof of Theorem 1,

$$R(f) \quad \leq \quad \hat{R}(f) + \sqrt{\frac{2\hat{V}_{out}(f)t}{m-1}} + \sum_{i=1}^{n}\sqrt{\frac{2m_i\hat{V}_{in}(f|e_i)t}{m(m-1)}} + \frac{2(4m-1)Mt}{3m(m-1)},$$

holds with probability larger than $1 - 2\exp(-t)$. Next, we consider a set of functions $\{f^1, \ldots, f^{N_\epsilon}\}$, which is a minimal $\epsilon$-cover of the function space $\mathcal{F}$ of size $N_\epsilon = N(\mathcal{F}, \epsilon, \|\cdot\|_{L^\infty(\mathcal{X} \times \mathcal{Y})})$. Then, for any $f \in \mathcal{F}$, there exists $f^j, 1 \leq j \leq N_\epsilon$ such that $\|f - f^j\|_{L^\infty(\mathcal{X} \times \mathcal{Y})} \leq \epsilon$. Therefore,

$$R(f) \quad \leq \quad R(f^j) + \epsilon$$

$$\leq \quad \hat{R}(f^j) + \sqrt{\frac{2\hat{V}_{out}(f^j)t}{m-1}} + \sum_{i=1}^{n}\sqrt{\frac{2m_i\hat{V}_{in}(f^j|e_i)t}{m(m-1)}}$$

$$+\frac{2(4m-1)Mt}{3m(m-1)} + \epsilon,$$

with probability larger than $1 - (2+n)\exp(-t)$. Notice that $\hat{R}(f^j) \leq \hat{R}(f) + \epsilon$ and

$$\sqrt{\hat{V}_{out}(f^j)} \quad \leq \quad \sqrt{\hat{V}_{out}(f)} + \sqrt{\hat{V}_{out}(f^j - f)} \leq \sqrt{\hat{V}_{out}(f)} + \epsilon,$$

$$\sqrt{\hat{V}_{in}(f^j|e_i)} \quad \leq \quad \sqrt{\hat{V}_{in}(f|e_i)} + \sqrt{\hat{V}_{in}(f^j - f|e_i)} \leq \sqrt{\hat{V}_{in}(f|e_i)} + \epsilon.$$

Therefore, for every $f \in \mathcal{F}$,

$$R(f) \quad \leq \quad \hat{R}(f) + \sqrt{\frac{2\hat{V}_{out}(f)t}{m-1}} + \sum_{i=1}^{n}\sqrt{\frac{2m_i\hat{V}_{in}(f|e_i)t}{m(m-1)}} \tag{21}$$

$$+\frac{2(4m-1)Mt}{3m(m-1)} + \Big(2 + \sqrt{\frac{2t}{m-1}} + \sum_{i=1}^{n}\sqrt{\frac{2m_it}{m(m-1)}}\Big)\epsilon,$$

holds with probability at least $1 - 2N_\epsilon\exp(-t)$. Since

$$\hat{f} \quad = \quad \arg\min_{f \in \mathcal{F}} \hat{R}(f) + \sqrt{\frac{2\hat{V}_{out}(f)t}{m-1}},$$

then we have

$$\hat{R}(\hat{f}) + \sqrt{\frac{2\hat{V}_{out}(\hat{f})t}{m-1}} \quad \leq \quad \hat{R}(f^*) + \sqrt{\frac{2\hat{V}_{out}(f^*)t}{m-1}}.$$

By the Bernstein inequality, with probability at least $1 - \exp(-t)$,

$$\hat{R}(f^*) \quad \leq \quad R(f^*) + \sqrt{\frac{2V(f^*)t}{m}} + \frac{2Mt}{3m}. \tag{22}$$

By Theorem 10 in [27],

$$\sqrt{\hat{V}_{out}(f^*)} \quad \leq \quad \sqrt{\hat{V}(f^*)} \leq \sqrt{\frac{m-1}{m}V(f^*)} + \sqrt{\frac{2M^2t}{m-1}}, \tag{23}$$

holds with probability larger than $1 - \exp(-t)$. Combining (21), (22) and (23),

$$R(\hat{f}) \quad \leq \quad R(f^*) + 2\sqrt{\frac{2V(f^*)t}{m}} + \sum_{i=1}^{n}\sqrt{\frac{2m_i\hat{V}_{in}(\hat{f}|e_i)t}{m(m-1)}}$$

$$+\frac{2Mt}{3m} + \frac{2Mt}{m-1} + \frac{2(4m-1)Mt}{3m(m-1)}$$

$$+\Big(2 + \sqrt{\frac{2t}{m-1}} + \sum_{i=1}^{n}\sqrt{\frac{2m_it}{m(m-1)}}\Big)\epsilon,$$

holds with probability larger than $1 - (2N_\epsilon + 2)\exp(-t)$.

$$\square$$

 # E    Proof of Theorem 3

510 **Theorem** *Suppose the training dataset $\mathcal{D}$ and a function $f \in \mathcal{F}$ are given. Let $\rho_+$ be the largest*
511 *distance between $\boldsymbol{q} \in \mathcal{Q}_\alpha(\hat{\boldsymbol{q}}, +\infty)$ and*

$$\rho_- = \frac{\min_i (\alpha/\hat{q}_i - 1)^2 \hat{V}_{out}(f)}{2\big(\min_i \hat{R}(f|e_i) - \hat{R}(f)\big)^2}.$$

512 *Then we have*

$$\mathcal{L}(f; \rho_-) \leq \max_{\boldsymbol{q} \in \mathcal{Q}_{\alpha, +\infty}(\hat{\boldsymbol{q}})} \sum_{i=1}^n q_i \hat{R}(f|e_i) \leq \mathcal{L}(f; \rho_+).$$

513 In this section, we decompose the complete proof into the three steps.

514 **Step 1.** Given a training dataset and a function $f \in \mathcal{F}$, the following inequality always holds:

$$\max_{\boldsymbol{q} \in \mathcal{Q}_{\alpha, \rho}(\hat{\boldsymbol{q}})} \sum_{i=1}^n q_i \hat{R}(f|e_i) \quad \leq \quad \hat{R}(f) + \sqrt{2\rho \hat{V}_{out}(f)}. \tag{24}$$

515 **Proof**: Since $\sum_{i=1}^n q_i = 1$ and $\sum_{i=1}^n \hat{q}_i = 1$, we have

$$\sum_{i=1}^n q_i \hat{R}(f|e_i) \quad = \quad \sum_{i=1}^n (q_i - \hat{q}_i) \hat{R}(f|e_i) + \sum_{i=1}^n \hat{q}_i \hat{R}(f|e_i)$$

$$= \quad \hat{R}(f) + \sum_{i=1}^n (q_i - \hat{q}_i) \hat{R}(f|e_i).$$

516 Since $\sum_{i=1}^n (q_i - \hat{q}_i) C = 0$ holds for any constant $C$,

$$\sum_{i=1}^n (q_i - \hat{q}_i) \hat{R}(f|e_i) \quad = \quad \sum_{i=1}^n (q_i - \hat{q}_i)\big(\hat{R}(f|e_i) - \hat{R}(f)\big).$$

517 Thus the max problem in (24) is equivalent to maximize

$$\max_{\boldsymbol{q} \in \mathcal{Q}_{\alpha, \rho}(\hat{\boldsymbol{q}})} \sum_{i=1}^n (q_i - \hat{q}_i)\big(\hat{R}(f|e_i) - \hat{R}(f)\big).$$

518 By the Cauchy-Schwarz inequality,

$$\sum_{i=1}^n (q_i - \hat{q}_i)\big(\hat{R}(f|e_i) - \hat{R}(f)\big)$$

$$= \quad \sum_{i=1}^n \frac{q_i - \hat{q}_i}{\sqrt{\hat{q}_i}} \sqrt{\hat{q}_i}\big(\hat{R}(f|e_i) - \hat{R}(f)\big)$$

$$\leq \quad \sqrt{\sum_{i=1}^n \frac{(q_i - \hat{q}_i)^2}{\hat{q}_i}} \times \sqrt{\sum_{i=1}^n \hat{q}_i\big(\hat{R}(f|e_i) - \hat{R}(f)\big)^2}$$

$$\leq \quad \sqrt{2\rho} \times \sqrt{\hat{V}_{out}(f)}$$

519                                                                                          $\square$

520 **Step 2.** If the between-domain variance $\hat{V}_{out}(f)$ is non-zero, the equality holds if and only if
521 $\forall e_i \in \mathcal{E}_{tr}$,

$$\alpha \leq \hat{q}_i \left( \sqrt{\frac{2\rho}{\hat{V}_{out}(f)}} \big(\hat{R}(f|e_i) - \hat{R}(f)\big) + 1 \right).$$

On the other hand, if $\alpha$ is fixed, the equality holds when the radius of $\mathcal{Q}_{\alpha,\rho}(\hat{q})$ satisfies

$$\rho \leq \frac{(\alpha/\hat{q}_i - 1)^2 \hat{V}_{out}(f)}{2\big(\min_i \hat{R}(f|e_i) - \hat{R}(f)\big)^2}.$$

**Proof**: The equality in (24) is attained if and only if the following requirements hold at the same time:

(i) There exists a constant $c$ such that $\forall 1 \leq i \leq n$,

$$\frac{q_i - \hat{q}_i}{\sqrt{\hat{q}_i}} = c\sqrt{\hat{q}_i}\big(\hat{R}(f|e_i) - \hat{R}(f)\big).$$

(ii) The $\chi^2$ divergence between $q$ and $\hat{q}$ achieves $\rho$, that is

$$\sum_{i=1}^{n} \frac{(q_i - \hat{q}_i)^2}{\hat{q}_i} = 2\rho.$$

It is easy to see

$$c^2 \sum_{i=1}^{n} \hat{q}_i \big(\hat{R}(f|e_i) - \hat{R}(f)\big)^2 = 2\rho \quad \Rightarrow \quad c = \sqrt{\frac{2\rho}{\hat{V}_{out}(f)}}.$$

Then the discrete distribution $q$ satisfies (i) is

$$q_i = \sqrt{\frac{2\rho}{\hat{V}_{out}(f)}} \hat{q}_i \big(\hat{R}(f|e_i) - \hat{R}(f)\big) + \hat{q}_i.$$

Since $q$ belongs to $\mathcal{Q}_{\alpha,\rho}(\hat{q})$, the only constraint here is $q_i \geq \alpha$, $\forall e_i$ which holds if and only if $\forall e_i \in \mathcal{E}_{tr}$,

$$\alpha \leq \hat{q}_i \Big(\sqrt{\frac{2\rho}{\hat{V}_{out}(f)}}\big(\hat{R}(f|e_i) - \hat{R}(f)\big) + 1\Big).$$

On the other hand, if $\alpha$ is fixed and non-positive, the constraint implies that the radius $\rho$ of $\mathcal{Q}_{\alpha,\rho}(\hat{q})$ should be sufficiently small:

$$\alpha \leq \hat{q}_i \Big(\sqrt{\frac{2\rho}{\hat{V}_{out}(f)}}\big(\hat{R}(f|e_i) - \hat{R}(f)\big) + 1\Big),$$

$$\Leftrightarrow \quad \frac{\alpha}{\hat{q}_i} - 1 \leq \sqrt{\frac{2\rho}{\hat{V}_{out}(f)}}\big(\hat{R}(f|e_i) - \hat{R}(f)\big),$$

$$\Leftarrow \quad \frac{\alpha/\hat{q}_i - 1}{\min_i \big(\hat{R}(f|e_i) - \hat{R}(f)\big)} \geq \sqrt{\frac{2\rho}{\hat{V}_{out}(f)}},$$

$$\Leftarrow \quad \rho \leq \frac{\min_i (\alpha/\hat{q}_i - 1)^2 \hat{V}_{out}(f)}{2\big(\min_i \hat{R}(f|e_i) - \hat{R}(f)\big)^2}.$$

Hence the proof is finished.

$\qquad\qquad\qquad\qquad\qquad\qquad\qquad\qquad\qquad\qquad\qquad\qquad\qquad\qquad\qquad\qquad\qquad\qquad\qquad\qquad$ $\square$

**Step 3.** Proof of (13) in Theorem 3.

**Proof**: First, $\rho_+$ is the largest distance between $\hat{q}$ and $q \in \mathcal{Q}_{\alpha,+\infty}(\hat{q})$. Therefore,

$$\mathcal{Q}_{\alpha,+\infty}(\hat{q}) = \mathcal{Q}_{\alpha,\rho_+}(\hat{q}) \subseteq \mathcal{Q}_{+\infty,\rho_+}(\hat{q}).$$

Hence the second inequality in (13) is trivial. Let $q^*$ be the solution:

$$q_-^* = \underset{q \in \mathcal{Q}_{+\infty,\rho_-}(\hat{q})}{\arg\max} \sum_{i=1}^{n} q_i \hat{R}(f|e_i),$$

According to **Step 2**,

$$\max_{\boldsymbol{q}\in\mathcal{Q}_{+\infty,\rho_-}(\hat{\boldsymbol{q}})}\sum_{i=1}^{n}q_i\hat{R}(f|e_i)=\mathcal{L}(f;\rho_-),$$

and

$$\boldsymbol{q}_-^*\in\mathcal{Q}_{+\infty,\rho_-}(\hat{\boldsymbol{q}})=\mathcal{Q}_{\alpha,\rho_-}(\hat{\boldsymbol{q}})\subseteq\mathcal{Q}_{\alpha,+\infty}(\hat{\boldsymbol{q}}).$$

Hence the first inequality in (13) is proved.

□

# F   Proof of Theorem 4

In this section, we start with a preliminary result.

**Theorem 9** *Suppose that $\alpha$ is a non-positive scalar and*

$$V'_{out}(f)=V(f)-\sum_{i=1}^{n}q_iV_{in}(f|e_i)>0.$$

*For each training domain, both $\hat{q}_i$ and $q_i$ are larger than $\delta>0$. Let*

$$\rho'=\frac{V'_{out}(f)}{8M^2}\left(\frac{\alpha}{1-(n-1)\delta}-1\right)^2.$$

*If $n>2$ and $m$ is sufficiently large such that*

$$\frac{m}{4}V'_{out}(f)>V(f),$$

*then, given $f\in\mathcal{F}$, the following expansion uniformly holds:*

$$\max_{\boldsymbol{q}\in\mathcal{Q}_{\alpha,\rho'}(\hat{\boldsymbol{q}})}\sum_{i=1}^{n}q_i\hat{R}(f|e_i)=\mathcal{L}(f;\rho'),$$

*with probability at least*

$$1-\exp\left(-\frac{\left(\frac{m}{4}V'_{out}(f)-V(f)\right)^2}{2M^2(m-1)V(f)}\right)-\binom{m+n-1}{n-1}\exp\left(-\frac{mV'_{out}(f)^2}{M^4}\right)$$
$$-\sum_{i=1}^{n}\exp\left(-\frac{(V_{in}(f|e_i)+\frac{m_i}{4}V'_{out}(f))^2}{2M^2m_i(V_{in}(f|e_i)+\frac{1}{4}V'_{out}(f))^2}\right).$$

**Proof**: Note that $\hat{R}(f|e)\in[0,M]$ for any $f\in\mathcal{F}$ and $e\in\mathcal{E}_{tr}$. Thus for any training data $\mathbb{D}$,

$$\left(\min_i\hat{R}(f|e_i)-\hat{R}(f)\right)^2\le M^2.$$

In addition, since $\alpha\le0$ and $q_i,\hat{q}_i\ge\delta$,

$$\min_i\left(\frac{\alpha}{\hat{q}_i}-1\right)^2\le\left(\frac{\alpha}{1-(n-1)\delta}-1\right)^2.$$

Hence, to satisfying

$$\rho'\le\frac{\min_i(\alpha/\hat{q}_i-1)^2\hat{V}_{out}(f)}{2\left(\min_i\hat{R}(f|e_i)-\hat{R}(f)\right)^2},$$

it suffices to show that

$$\hat{V}_{out}(f)\ge\frac{1}{4}V'_{out}(f).$$

Notice that

$$\hat{V}_{out}(f) = \hat{V}(f) - \sum_{i=1}^{n} \hat{q}_i \hat{V}_{in}(f|e_i) = V'_{out}(f) + I_1 + I_2 + I_3,$$

where

$$
\begin{aligned}
I_1 &= \hat{V}(f) - V(f), \\
I_2 &= -\sum_{i=1}^{n} (\hat{q}_i - q_i) \hat{V}_{in}(f|e_i), \\
I_3 &= -\sum_{i=1}^{n} q_i \big( \hat{V}_{in}(f|e_i) - V_{in}(f|e_i) \big).
\end{aligned}
$$

By Theorem 10 in [27], for any $\delta > 0$,

$$P\Big( \frac{m}{m-1} \hat{V}(f) < V(f) - \delta \Big) \le \exp\Big( -\frac{(m-1)\delta^2}{2M^2 V(f)} \Big).$$

We take

$$\delta = \frac{m}{4(m-1)} V'_{out}(f) - \frac{1}{m-1} V(f).$$

Therefore,

$$
\begin{aligned}
P\Big( I_1 < -\frac{1}{4} V'_{out}(f) \Big) &= P\Big( \hat{V}(f) < V(f) - \frac{1}{4} V'_{out}(f) \Big) \\
&\le \exp\Big( -\frac{\big( \frac{m}{4} V'_{out}(f) - V(f) \big)^2}{2M^2(m-1)V(f)} \Big).
\end{aligned}
$$

For the term $I_2$,

$$|I_2| \le \sum_{i=1}^{n} |\hat{q}_i - q_i| \hat{V}_{in}(f|e_i) \le \|\hat{q} - q\|_1 \frac{M^2}{4}.$$

According to the Pinkser's inequality and the method of types [15], for any $\delta > 0$,

$$\|\hat{q} - q\|_1 \le \sqrt{2D_{KL}(\hat{q}\|q)} < \delta$$

with probability at least

$$1 - \binom{m+n-1}{n-1} \exp\Big( -m\delta^2 \Big).$$

We take $\delta = V'_{out}(f)/M^2$. Then we know

$$P\Big( I_2 < -\frac{1}{4} V'_{out}(f) \Big) \le \binom{m+n-1}{n-1} \exp\Big( -\frac{mV'_{out}(f)^2}{M^4} \Big).$$

Next, we deal with $I_3$. By Theorem 10 in [27], for any $\delta > 0$ and any $e_i \in \mathcal{E}_{tr}$,

$$P\Big( \frac{m}{m-1} \hat{V}_{in}(f|e_i) > V_{in}(f|e_i) + \delta \Big) \le \exp\Big( -\frac{(m_i-1)\delta^2}{2M^2(V_{in}(f|e_i)+\delta)} \Big).$$

Let

$$\delta = \frac{1}{m_i-1} V_{in}(f|e_i) + \frac{m_i}{4(m_i-1)} V'_{out}(f).$$

Then we have

$$
\begin{aligned}
P\Big( \hat{V}_{in}(f|e_i) > V_{in}(f|e_i) + \frac{1}{4} V'_{out}(f) \Big) &= P\Big( \frac{m}{m-1} \hat{V}_{in}(f|e_i) > V_{in}(f|e_i) + \delta \Big) \\
&\le \exp\Big( -\frac{(V_{in}(f|e_i) + \frac{m_i}{4} V'_{out}(f))^2}{2M^2 m_i (V_{in}(f|e_i) + \frac{1}{4} V'_{out}(f))^2} \Big).
\end{aligned}
$$

564 Therefore $I_3 < -\frac{1}{4}V'_{out}(f)$ with probability smaller thant

$$\sum_{i=1}^{n} \exp\left(-\frac{(V_{in}(f|e_i) + \frac{m_i}{4}V'_{out}(f))^2}{2M^2 m_i(V_{in}(f|e_i) + \frac{1}{4}V'_{out}(f))^2}\right)$$

565 Combining the results of $I_1$, $I_2$ and $I_3$, we have

$$\hat{V}_{out}(f) \geq \frac{1}{4}V'_{out}(f),$$

566 with probability larger than

$$1 - \exp\left(-\frac{\left(\frac{m}{4}V'_{out}(f) - V(f)\right)^2}{2M^2(m-1)V(f)}\right) - \binom{m+n-1}{n-1}\exp\left(-\frac{mV'_{out}(f)^2}{M^4}\right)$$

$$- \sum_{i=1}^{n}\exp\left(-\frac{(V_{in}(f|e_i) + \frac{m_i}{4}V'_{out}(f))^2}{2M^2 m_i(V_{in}(f|e_i) + \frac{1}{4}V'_{out}(f))^2}\right).$$

567 Hence the proof is finished.

568 □

569 Next, we extend Theorem 9 to a more general variant with respect to the family of functions $\mathcal{F}$.

570 **Theorem.** *Suppose that $\alpha$ is a non-positive scalar and $V'_{out}(f) = V(f) - \sum_i q_i V_{in}(f|e_i) > 0$. For*
571 *each training domain, both $\hat{q}_i$ and $q_i$ are larger than $\delta > 0$. We write*

$$\rho' = \frac{V'_{out}(f)}{16M^2}\left(\frac{\alpha}{1 - (n-1)\delta} - 1\right)^2.$$

572 *Let $\tau > 0$ and $0 < \eta < 1$ be two constants. Define*

$$\mathcal{F}_{\tau,\eta} = \{f \in \mathcal{F} : V(f) \geq \tau, \text{ and } \frac{V_{in}(f|e_i)}{V(f)} \leq \eta, \forall e_i \in \mathcal{E}_{tr}\}.$$

573 *For any $f \in \mathcal{F}_{\tau,\eta}$, the following expansion uniformly holds:*

$$\max_{q \in \mathcal{Q}_{\alpha,\rho'}(\hat{q})} \sum_{i=1}^{n} q_i \hat{R}(f|e_i) = \mathcal{L}(f; \rho'),$$

574 *with probability at least $1 - N_{\tau,\eta} \times p$, where $N_{\tau,\eta} = N\left(\mathcal{F}_{\tau,\eta}, \sqrt{\frac{1}{10}(1-\eta)\tau}, \|\cdot\|_{L^\infty(\mathcal{X}\times\mathcal{Y})}\right)$ is the*
575 *covering number of $\mathcal{F}_{\tau,\eta}$ and*

$$p = \exp\left(-\frac{m(1-\eta)^2\tau}{32M^2} + \frac{1}{16}\right) + \binom{m+n-1}{n-1}\exp\left(-\frac{m(1-\eta)^2\tau^2}{M^4}\right)$$

$$+ \sum_{i=1}^{n}\exp\left\{-\frac{1}{2M^2 m_i}\left(\frac{m_i(1-\eta) + 4\eta}{1+3\eta}\right)^2\right\}.$$

576 **Proof**: We consider a set of functions $\{f^1, \ldots, f^N\}$, which is a minimal $\epsilon$-cover of $\mathcal{F}_{\tau,\eta}$ of size

$$N_{\tau,\eta} = N(\mathcal{F}_{\tau,\eta}, \epsilon, \|\cdot\|_{L^\infty(\mathcal{X}\times\mathcal{Y})}).$$

577 Define the event

$$\mathfrak{E} = \left\{\hat{V}_{out}(f^i) \geq \frac{1}{4}V'_{out}(f^i), \text{ for } i = 1, \ldots, N\right\}.$$

578 Recall the proof of Theorem 3 and the terms $I_1$ to $I_3$. For any $f \in \mathcal{F}_{\tau,\eta}$,

$$P\left(I_1 < -\frac{1}{4}V'_{out}(f)\right) \leq \exp\left(-\frac{\left(\frac{m}{4}V'_{out}(f) - V(f)\right)^2}{2M^2(m-1)V(f)}\right)$$

$$= \exp\left(-\frac{m^2 V'_{out}(f)^2}{32M^2(m-1)V(f)} + \frac{mV'_{out}(f)}{4M^2(m-1)} - \frac{V(f)}{2M^2(m-1)}\right)$$

$$\leq \exp\left(-\frac{m^2 V'_{out}(f)^2}{32M^2(m-1)V(f)} + \frac{1}{16}\right)$$

$$\leq \exp\left(-\frac{m(1-\eta)^2\tau}{32M^2} + \frac{1}{16}\right).$$

579 For the term $I_2$,

$$P\Big(I_2 < -\frac{1}{4}V'_{out}(f)\Big) \leq \left(\begin{array}{c} m+n-1 \\ n-1 \end{array}\right) \exp\Big(-\frac{mV'_{out}(f)^2}{M^4}\Big)$$

$$\leq \left(\begin{array}{c} m+n-1 \\ n-1 \end{array}\right) \exp\Big(-\frac{m(1-\eta)^2\tau^2}{M^4}\Big).$$

580 For any $e_i \in \mathcal{E}_{tr}$,

$$-\frac{(V_{in}(f|e_i) + \frac{m_i}{4}V'_{out}(f))^2}{2M^2m_i(V_{in}(f|e_i) + \frac{1}{4}V'_{out}(f))^2} = -\frac{1}{2M^2m_i}\Big(\frac{\frac{m_i-1}{4}V'_{out}(f)}{V_{in}(f|e_i) + \frac{1}{4}V'_{out}(f)} + 1\Big)^2$$

$$= -\frac{1}{2M^2m_i}\Big(\frac{m_i-1}{4\frac{V_{in}(f|e_i)}{V'_{out}(f)} + 1} + 1\Big)^2$$

$$\leq -\frac{1}{2M^2m_i}\Big(\frac{m_i-1}{4\frac{\eta}{1-\eta} + 1} + 1\Big)^2$$

$$= -\frac{1}{2M^2m_i}\Big(\frac{(m_i-1)(1-\eta)}{1+3\eta} + 1\Big)^2$$

$$= -\frac{1}{2M^2m_i}\Big(\frac{m_i(1-\eta)+4\eta}{1+3\eta}\Big)^2$$

581 Therefore, for the term $I_3$,

$$P\Big(I_3 < -\frac{1}{4}V'_{out}(f)\Big) \leq \sum_{i=1}^n \exp\Big\{-\frac{1}{2M^2m_i}\Big(\frac{m_i(1-\eta)+4\eta}{1+3\eta}\Big)^2\Big\}.$$

582 Furthermore, we have for any $f \in \mathcal{F}_{\tau,\eta}$,

$$\hat{V}_{out}(f) \geq \frac{1}{4}V'_{out}(f),$$

583 with probability larger than $1-p$, where

$$p = \exp\Big(-\frac{m(1-\eta)^2\tau}{32M^2} + \frac{1}{16}\Big) + \left(\begin{array}{c} m+n-1 \\ n-1 \end{array}\right) \exp\Big(-\frac{m(1-\eta)^2\tau^2}{M^4}\Big)$$

$$+ \sum_{i=1}^n \exp\Big\{-\frac{1}{2M^2m_i}\Big(\frac{m_i(1-\eta)+4\eta}{1+3\eta}\Big)^2\Big\}.$$

584 Then, combining the results of $\{f^j, j = 1, \ldots, N\}$,

$$P(\mathfrak{E}) \geq 1 - N(\mathcal{F}_{\tau,\eta}, \epsilon, \|\cdot\|_{L^\infty(\mathcal{X}\times\mathcal{Y})}) \times p.$$

Given the event $\mathfrak{E}$, for any $f \in \mathcal{F}$, there exists $f^j$ such that

$$\|f - f^j\|_{L^\infty(\mathcal{X}\times\mathcal{Y})} \leq \epsilon.$$

585 Hence,

$$\hat{V}_{out}(f) \geq \hat{V}_{out}(f^j) - \epsilon^2$$

$$\geq \frac{1}{4}V'_{out}(f^j) - \epsilon^2 \geq \frac{1}{4}V'_{out}(f) - \frac{5}{4}\epsilon^2.$$

586 We take

$$\epsilon = \sqrt{\frac{1}{10}(1-\eta)\tau}.$$

587 Then,

$$\hat{V}_{out}(f) \geq \frac{1}{4}V'_{out}(f) - \frac{5}{4}\epsilon^2 \geq \frac{1}{8}V'_{out}(f).$$

588 Hence the proof is finished.

589 □

 # G  Proof of Theorem 5

591 **Theorem.** *Given the training dataset and a function $f \in \mathcal{F}$, then for any anchor distribution $\boldsymbol{q}_0$, the*
592 *inequality always holds:*

$$\max_{\boldsymbol{q} \in \mathcal{Q}_\alpha(\boldsymbol{q}_0, \rho)} \sum_{i=1}^n q_i \hat{R}(f|e_i) \leq \hat{R}(f, \boldsymbol{q}_0) + \sqrt{2\rho \hat{V}_{out}(f, \boldsymbol{q}_0)},$$

593 *where*

$$\hat{R}(f, \boldsymbol{q}_0) = \frac{1}{n} \sum_{i=1}^n q_{0,i} \hat{R}(f|e_i),$$

$$\hat{V}_{out}(f, \boldsymbol{q}_0) = \sum_{i=1}^n q_{0,i} \big(\hat{R}(f|e_i) - \hat{R}(f)\big)^2.$$

594 *If the between-domain variance $\hat{V}_{out}(f, \boldsymbol{q}_0)$ is non-zero, the equality holds if and only if $\forall e_i \in \mathcal{E}_{tr}$,*

$$\alpha \leq q_{0,i} \left( \sqrt{\frac{2\rho}{\hat{V}_{out}(f, \boldsymbol{q}_0)}} \big(\hat{R}(f|e_i) - \hat{R}(f, \boldsymbol{q}_0)\big) + 1 \right).$$

595 *On the other hand, if $\alpha$ is fixed, the equality holds when the radius of $\mathcal{Q}_\alpha(\boldsymbol{q}_0, \rho)$ satisfies,*

$$\rho \leq \frac{\min_i (\alpha/q_{0,i} - 1)^2 \hat{V}_{out}(f, \boldsymbol{q}_0)}{2 \big( \min_i \hat{R}(f|e_i) - \hat{R}(f, \boldsymbol{q}_0)\big)^2}.$$

596 *The proof here is similar to that of Theorem 3.*

597 **Proof**: Since $\sum_{i=1}^n q_i = 1$ and $\sum_{i=1}^n q_{0,i} = 1$, we have

$$\sum_{i=1}^n q_i \hat{R}(f|e_i) = \sum_{i=1}^n q_{0,i} \hat{R}(f|e_i) + \sum_{i=1}^n (q_i - q_{0,i}) \hat{R}(f|e_i)$$

$$= \hat{R}(f, \boldsymbol{q}_0) + \sum_{i=1}^n (q_i - q_{0,i}) \hat{R}(f|e_i).$$

598 Since $\sum_{i=1}^n (q_i - \hat{q}_i) = 0$, then we have

$$\sum_{i=1}^n (q_i - q_{0,i}) \hat{R}(f|e_i) = \sum_{i=1}^n (q_i - q_{0,i}) \big(\hat{R}(f|e_i) - \hat{R}(f, \boldsymbol{q}_0)\big).$$

599 Thus the max problem with respect to $\boldsymbol{q}$ is equivalent to maximize

$$\max_{\boldsymbol{q} \in \mathcal{Q}_\alpha(\boldsymbol{q}_0, \rho)} \sum_{i=1}^n (q_i - q_{0,i}) \big(\hat{R}(f|e_i) - \hat{R}(f, \boldsymbol{q}_0)\big).$$

600 By the Cauchy-Schwarz inequality,

$$\sum_{i=1}^n (q_i - q_{0,i}) \big(\hat{R}(f|e_i) - \hat{R}(f, \boldsymbol{q}_0)\big)$$

$$= \sum_{i=1}^n \frac{q_i - q_{0,i}}{\sqrt{q_{0,i}}} \sqrt{q_{0,i}} \big(\hat{R}(f|e_i) - \hat{R}(f, \boldsymbol{q}_0)\big)$$

$$\leq \sqrt{\sum_{i=1}^n \frac{(q_i - q_{0,i})^2}{q_{0,i}}} \times \sqrt{\sum_{i=1}^n q_{0,i} \big(\hat{R}(f|e_i) - \hat{R}(f, \boldsymbol{q}_0)\big)^2}$$

$$\leq \sqrt{2\rho} \times \sqrt{\hat{V}_{out}(f, \boldsymbol{q}_0)}$$

601 The equality is attained if and only if the following requirements hold at the same time:

(i) There exists a constant $c$ such that $\forall 1 \leq i \leq n$,

$$\frac{q_i - q_{0,i}}{\sqrt{q_{0,i}}} = c\sqrt{q_{0,i}}\big(\hat{R}(f|e_i) - \hat{R}(f, \boldsymbol{q}_0)\big).$$

(ii) The $\chi^2$ divergence between $\boldsymbol{q}$ and $\boldsymbol{q}_0$ achieves $\rho$:

$$\sum_{i=1}^{n} \frac{(q_i - q_{0,i})^2}{q_{0,i}} = 2\rho.$$

It is easy to see

$$c^2 \sum_{i=1}^{n} q_{0,i}\big(\hat{R}(f|e_i) - \hat{R}(f, \boldsymbol{q}_0)\big)^2 = 2\rho \quad \Rightarrow \quad c = \sqrt{\frac{2\rho}{\hat{V}_{out}(f, \boldsymbol{q}_0)}}.$$

Then the discrete distribution $\boldsymbol{q}$ satisfies (i) is

$$q_i = \sqrt{\frac{2\rho}{\hat{V}_{out}(f, \boldsymbol{q}_0)}} q_{0,i}\big(\hat{R}(f|e_i) - \hat{R}(f, \boldsymbol{q}_0)\big) + q_{0,i}.$$

Since $\boldsymbol{q}$ belongs to $\mathcal{Q}_\alpha(\boldsymbol{q}_0, \rho)$, the constraint here is $q_i \geq \alpha$, $\forall e_i$ which holds if and only if

$$\alpha \leq q_{0,i}\Big(\sqrt{\frac{2\rho}{\hat{V}_{out}(f)}}\big(\hat{R}(f|e_i) - \hat{R}(f)\big) + 1\Big), \ \forall e_i \in \mathcal{E}_{tr}.$$

On the other hand, if $\alpha$ is fixed and non-positive, the constraint implies that the radius $\rho$ of $\mathcal{Q}_\alpha(\boldsymbol{q}_0, \rho)$ should be sufficiently small:

$$\alpha \leq q_{0,i}\Big(\sqrt{\frac{2\rho}{\hat{V}_{out}(f, \boldsymbol{q}_0)}}\big(\hat{R}(f|e_i) - \hat{R}(f, \boldsymbol{q}_0)\big) + 1\Big)$$

$$\Leftrightarrow \quad \frac{\alpha}{q_{0,i}} - 1 \leq \sqrt{\frac{2\rho}{\hat{V}_{out}(f, \boldsymbol{q}_0)}}\big(\hat{R}(f|e_i) - \hat{R}(f, \boldsymbol{q}_0)\big),$$

$$\Leftarrow \quad \frac{\alpha/q_{0,i} - 1}{\min_i \big(\hat{R}(f|e_i) - \hat{R}(f, \boldsymbol{q}_0)\big)} \geq \sqrt{\frac{2\rho}{\hat{V}_{out}(f, \boldsymbol{q}_0)}},$$

$$\Leftarrow \quad \rho \leq \frac{\min_i(\alpha/q_{0,i} - 1)^2 \hat{V}_{out}(f, \boldsymbol{q}_0)}{2\big(\min_i \hat{R}(f|e_i) - \hat{R}(f, \boldsymbol{q}_0)\big)^2}.$$

Hence the proof is finished.

$\square$