# OpenReview forum: "Can Variance-Based Regularization Improve Domain Generalization?"
_NeurIPS.cc/2022/Conference — NeurIPS 2022 Submitted_

### Official Review · Reviewer_QGcy · 2022-07-10

**Rating:** 5
**Confidence:** 2
**Soundness:** 4 excellent
**Presentation:** 3 good
**Contribution:** 3 good

**Summary:**

In this paper, the author studied the problem of whether the variance-based regularization approach can improve domain generalization. With a mild assumption of the out-of-distribution data is generated by the shift of the domain distribution, the paper proposed a weighting correction scheme based on the previous variance-regularized domain generalization methods. They prove the guarantees for in-distribution generalization and show the potential advantages of the proposed method when compared to the Empirical Risk Minimization. Experimental results show improvements in some cases.

**Questions:**


1) Although the proposed method has a different assumption compared to invariant learning methods, is there any connections between these two approaches?

2) In Line 18-20, do you assum that the OOD test data is generated by domain shift from one single domain or a mixture of multiple domains?


**Strengths And Weaknesses:**

Strength

1) The paper is well-written with strong mathematical support.

2) The paper studies the question of whether variance-based regularization can improve domain generalization. Based on strong theoretical support, they proposed a simple weighting correction scheme and provide guarantees of in-distribution generalization.

Weakness

1) It could be better to include the experiments in the main paper. The main paper did not mention that there are experiments in the Appendix. More experiments can be done to support the proposed idea.

2) Based on the description from Line 194 to 204, the sample size m is critical to the proposed approach. In Line 199, it looks like a larger m could benefit the proposed method while Line 203 shows that the m should not be too large.  An empirical experiment might be required to demonstrate the impact of the sample size m.

---

> ### Author Response · Authors · 2022-08-02
> **Reply to Reviewer QGcy**
>
> We thank you for your time and insightful questions. We are encouraged by your positive feedback. We answer your questions here.
>
>
> ‘**W**’ for Weakness, ‘**Q**’ for Question, and '**Ans**' for Answer.
>
> ---
>
> **W1.** It could be better to include the experiments in the main paper. The main paper did not mention that there are experiments in the Appendix. More experiments can be done to support the proposed idea.
>
> **Ans.** Thank you for the comments. Due to the space limitation, we postpone experiments and two theorems into the appendix. If the final revision, we will include the experiments in the additional content page and add experimental comparisons with ERM to validate the discussions after Corollary 2.
>
> ---
>
> **W2.** Based on the description from Line 194 to 204, the sample size m is critical to the proposed approach. In Line 199, it looks like a larger m could benefit the proposed method while Line 203 shows that the m should not be too large. An empirical experiment might be required to demonstrate the impact of the sample size m.
>
> **Ans.** Line 194-204 discuss the convergence rate of the excess risk as $m$ tends to infinity. The method with a faster convergence rate can outperform competitors when $m$ is sufficiently large.
>
> - At Line 194-199, we assume there exists an optimal solution $f* \in \arg\min_f R(f)$ such that $V(f*)=0$. In this case, the convergence rate of the proposed method is faster than that of ERM.
>
> - At Line 200-204, we assume there is no optimal function $f^* \in \arg\min_f R(f)$ satisfies $V(f^*)=0$. In this case, the convergence rate of the proposed method is slower than that of ERM.
>
> In summary, Line 194-199 have a different assumption than Line 200-204. So these two parts come to completely different conclusions.
>
> ---
>
> **Q1.** Although the proposed method has a different assumption compared to invariant learning methods, is there any connections between these two approaches?
>
> **Ans.** Thank you for the question. In Section 2.2, we show the connections between the proposed method and invariant learning. We reformulate an invariant learning task as a hypothesis testing problem and show the relationship between the regularization $\hat V_{out}(f,\hat q)$ and the hypothesis testing problem (Line 117-135).
>
> ---
>
> **Q2.** In Line 18-20, do you assume that the OOD test data is generated by domain shift from one single domain or a mixture of multiple domains?
>
> **Ans.** We assume that the OOD test data is generated from a mixture of multiple domains. The single-domain test data is a special case that can be obtained by changing the support set of the domain distribution.
>
> ---

---

> > ### Author Response · Authors · 2022-08-09
> > **Response to Reviewer QGcy**
> >
> > Hello Reviewer QGcy, we are grateful for your time and constructive suggestions. We hope that our responses have addressed your concerns. If there remain concerns or you have more questions, we would be more than happy to provide additional clarification.
> >
> > To recap our response,
> >
> > - Lines 194-204 discuss the rate of convergence of excess risk as m tends to infinity. Lines 194-199 have different assumptions than lines 200-204. So the two parts come to completely different conclusions.
> >
> > - We show the connections between the proposed method and invariant learning (risk invariance) via hypothesis testing.
> >
> > - Single-domain test data is a special case of the mixture of multiple domains.

---

### Official Review · Reviewer_eL3F · 2022-07-11

**Rating:** 4
**Confidence:** 4
**Soundness:** 2 fair
**Presentation:** 2 fair
**Contribution:** 3 good

**Summary:**

This paper addresses the connection between domain generalization and group distributionally robust optimization. Variance-regualarized ERM is approximated by Group DRO with $\phi$-divergence ball. Unlike previous works, they take empirical domain distribution as an anchor of uncertainty sets of DRO instead of the uniform distribution, and provides guarantees of in-distribution generalization. They argue that with some conditions, their method can outperform ERM for domain generalization.

**Questions:**

I cannot fully understand the benefits of using empirical domain distribution as a anchor distribution of $phi$-divergence ball, instead of the uniform distribution. can it be  shown through some experiments?

**Limitations:**

The author have addressed the limitations.

**Strengths And Weaknesses:**

Strengths
* This paper did a focused job of generalizing and analysizing the connection between varizance-based regualarization and domain generalization. Their theoretical result for generalization bound for in-distribution seems solid.

Weaknesses
* Their work seems similar to [1]. The work [1] showed that group DRO formulation with unifrom distribution as a anchor is bounded with risk variance and the equality can be established with some condition. In this paper, the study is extended into general anchor including empirical distribution. However, theorem 3 and 4 can be straightforwardly extended from the results of [1].

* They argue that a weighting correction scheme can improve generalization in some cases. It would be better to show some experiments to make their arguments valid.

1. Xie, et al. Risk variance penelization, 2020.

---

> ### Author Response · Authors · 2022-08-02
> **Reply to Reviewer eL3F**
>
>
> Thank you for your time and and valuable feedback. We answer the questions here.
>
> ‘**W**’ for Weakness, ‘**Q**’ for Question, and '**Ans**' for Answer.
>
> ---
>
> **W1.** Their work seems similar to [1]. The work [1] showed that group DRO formulation with unifrom distribution as a anchor is bounded with risk variance and the equality can be established with some condition. In this paper, the study is extended into general anchor including empirical distribution. However, theorem 3 and 4 can be straightforwardly extended from the results of [1].
>
> **Ans.** Our work is very different from [1] in the following aspects:
>
> - **Data Generation.** The work [1] assumes a two-stage data generation. The first stage generates domains, and then the second stage generates the in-domain data points. Here we treat the domain label as an endogenous character and use a one-stage data generation process. Please see Line 21-23.
>
> - **Applicability.** Our setting is more suitable for explaining the existing variance-regularized methods. [1] have proposed the relationship between group DRO objective and the standard deviation of risk. But their results fail to explain the utility of penalizing risk variance (REx, [2]). Under our settings, we point out the utility of REx from the view of hypothesis testing (Section 2.2, Line 117 - Line 135).
>
> - **Theoretical Results.** [1] focuses on the optimization properties. In this work, we study both generalization and optimization. Our results show the benefits of the weighting correction and the risk of using variance-regularized methods. We also prove the optimization equivalence between group DRO and the proposed method.
>
> - **Technical Details.** Theorem 3 can be straightforwardly extended from the results of [1] (Proposition 2 in [1]). However, the technical details of our work are quite different from [1]. In the settings of [1], the two stages of data generation are disentangled. Therefore, the results in [1] (Theorem 3 and Theorem 4 in [1]) are derived by directly combining the inter-domain generalization error and the intra-domain sample-level generalization error. In our work, the data generation is one-stage. So our results (Theorem 1 and Theorem 4) are obtained by decomposing the risk variance (Line 152 - Line 153) and analyzing the generalization error of the pooled training data ($\hat R(f)$ and $R(f)$).
>
> ---
>
> **W2.** They argue that a weighting correction scheme can improve generalization in some cases. It would be better to show some experiments to make their arguments valid.
>
> **Ans.** Thank you for the comments. In Appendix A, we present ID experimental results to verify Theorem 1 (Line 164). Here we provide OOD results for the imbalance setting (Line 398-403):
>
> ***PACS***
> |method|A|C|P|S|Avg|
> |:---|----|----|----|----|----|
> Weighting Correction| 85.7 $\pm$ 1.0 | 80.3 $\pm$ 0.8 | 97.4 $\pm$ 0.2 | 77.8 $\pm$ 0.3 | 85.3 $\pm$ 0.6 |
> Standard Deviation | 47.8 $\pm$ 7.0 | 51.7 $\pm$ 1.9 | 91.7 $\pm$ 2.5 | 51.2 $\pm$ 5.5 | 60.6 $\pm$ 1.9 |
> Variance | 12.0 $\pm$ 0.6 | 18.4 $\pm$ 1.0 | 10.9 $\pm$ 1.1 | 20.3 $\pm$ 0.8 | 15.4  $\pm$ 0.4 |
>
> ***VLCS***
> |method|C|L|S|V|Avg|
> |:---|----|----|----|----|----|
> Weighting Correction| 98.2 $\pm$ 0.4 | 62.3 $\pm$ 0.8 | 71.2 $\pm$ 0.3 | 76.2 $\pm$ 1.3 | 77.0 $\pm$ 0.6 |
> Standard Deviation | 69.5 $\pm$ 0.3 | 47.2 $\pm$ 0.1 | 38.3 $\pm$ 0.5 | 46.3 $\pm$ 0.7 | 50.3 $\pm$ 0.1 |
> Variance | 58.8 $\pm$ 2.0 | 46.3 $\pm$ 0.2 | 38.4 $\pm$ 0.4 | 43.6 $\pm$ 0.7 | 46.8  $\pm$ 0.6 |
>
>
> The experimental results show that our weighting correction method provides a better OOD accuracy compared with other variance-regularized methods in the imbalance setting.
> We will cite the experimental evidence and explain the improvement of our method in the revision.
>
>
> ---
>
> **Q1.** I cannot fully understand the benefits of using empirical domain distribution as a anchor distribution of -divergence ball, instead of the uniform distribution. can it be shown through some experiments?
>
> **Ans.** Thank you for the question. In Appendix A, we present experimental results to validate our theoretical results. Please see our answer to **W2.**
>
> A heuristic understanding using the empirical domain distribution is as follows. Domain generalization is ultimately a generalization problem from training data to test data. The concept "domain" is a tool for describing the structure of training data. Using a uniform domain distribution already changes the distribution of the training data before dealing with domain generalization task. Changing the training distribution without prior information introduces risks to both the ID and OOD generalization. While using the empirical domain distribution can keep the distribution of training data.
>
> ---
>
> **Reference**
>
> [1] Xie, et al., Risk variance penelization, 2020.
>
> [2] Krueger, et al., Out-of-distribution generalization via risk extrapolation (rex). ICML 2021.

---

> > ### Comment · Reviewer_eL3F · 2022-08-07
> > **Reply to authors**
> >
> > Thank you for your all replies.
> >
> > I still have one complain and question for your work.
> >
> > * I agree that your work is different from [1] in that you showed the utility of risk variance regularization and the generalization bound by variances through Theorem 1. However, if Theorem 3 and 4 can be straightforwardly extended from the results of [1], as you mentioned in your reply, you should give appropriate credit to [1] in your paper. Otherwise, it may lead a reader that is not familiar with this area to believe that the idea of connection between risk variances and group DRO is fist in this paper.
> >
> > * Comparing Weighting correction and Variance with Balance, the reported results in Appendix are similar. (I compared your method to ``Variance`` with Balance because using Balance scheme is their own implementation.) So, I cannot still be confident that, in practice, the weighting correction scheme is indeed necessary for DG.

---

> > > ### Author Response · Authors · 2022-08-07
> > > **Thank you for your response.**
> > >
> > > Thank you for your response.
> > >
> > > ---
> > >
> > > - Re: "I agree that your work is different from [1] in that you showed the utility of risk variance regularization and the generalization bound by variances through Theorem 1. However, if Theorem 3 and 4 can be straightforwardly extended from the results of [1], as you mentioned in your reply, you should give appropriate credit to [1] in your paper. Otherwise, it may lead a reader that is not familiar with this area to believe that the idea of connection between risk variances and group DRO is fist in this paper."
> > >
> > > **Ans.** Thank you for the comments. We are sorry for the typo and have corrected it in our response. Technically, Theorem 3 **can** be extended from the results of [1] (Proposition 2 in [1]). Here Theorem 3 is a preliminary result for the main result.  Theorem 4 is the main result and **cannot** be extended from the results of [1] (Theorem 3 and 4 in [1]). The technical details behind Theorem 4 are very different from those in [1]. We will emphasize the connection and difference here in the revised version.
> > >
> > > ---
> > >
> > > - Re: "Comparing Weighting correction and Variance with Balance, the reported results in Appendix are similar. (I compared your method to Variance with Balance because using Balance scheme is their own implementation.) So, I cannot still be confident that, in practice, the weighting correction scheme is indeed necessary for DG."
> > >
> > > **Ans.**
> > >
> > > > Comparing Weighting correction and Variance with Balance, the reported results in Appendix are similar. (I compared your method to Variance with Balance because using Balance scheme is their own implementation.)
> > >
> > > In Appendix A, we have marked the key results in red and compared our method to Variance with Balance. Table 1 reports the accuracy and its standard deviation.
> > > With these results, we can find that (1) our method improves the ID generalization:   96.9 vs 96.7 and 84.7 vs 84.3; (2) The improvement of our method is statistically significant because the std of the accuracy is very small. For example, we can compute the 95% confidence interval of ID accuracy of PACS as [96.9 - 1.96 * 0.1, 96.9 + 1.96 * 0.1] and 96.7 does not fall in this interval. Therefore, at the significance level of 5% (i.e., the p-value is smaller than 0.05), our method improves the ID accuracy significantly.
> > >
> > > > So, I cannot still be confident that, in practice, the weighting correction scheme is indeed necessary for DG."
> > >
> > > The answer to this question needs to be discussed on a case-by-case basis.
> > >
> > > - In this work, we consider a standard setting: all we see in practice is the multi-sourced training data without prior information. In this case, an ID generalization guarantee is necessary and our method provides the guarantee for the variance-regularized method for DG (Line 173-184 and Line 205-213).
> > >
> > > - If we have strong prior information to restrict the target domain to a small range, then we can focus more on the target domain and less on the ID generalization problem. An extreme example is domain adaptation.
> > >
> > > ---
> > >
> > > **Reference:**
> > >
> > > [1] Xie, et al., Risk variance penalization, 2020.

---

> > > > ### Author Response · Authors · 2022-08-09
> > > > **Response to Reviewer eL3F**
> > > >
> > > > Hello Reviewer eL3F, we hope that we have already addressed your concerns. Please let us know if you still have other concerns.

---

> > > > ### Comment · Reviewer_eL3F · 2022-08-09
> > > > **Thank you for your reply!**
> > > >
> > > > Thank you for your detailed reply.
> > > >
> > > > I agree that your theoretical results, Theorem 1 and Theorem 4, are indeed your own contributions. (I request this will be emphasized in a revised version)
> > > >
> > > > However, I am still not sure about experimental results and the relationship between ID generalization and OOD generalization (pointed out by nws1). As you explained, your method outperformed two baselines on two datasets, but it seems marginal to me. Therefore, I think it would better to include more DG baselines and datasets to experimentally support your theoretical view. So, I am increasing my score to 4. I really appreciate your reply again.

---

> > > > > ### Author Response · Authors · 2022-08-09
> > > > > **Response to Reviewer eL3F**
> > > > >
> > > > > Thank you for raising the score. In the revision, we will emphasize the contribution of our work and highlight the novelty of the theoretical results. We will also consider more experiments or simulations to support our approach.

---

### Official Review · Reviewer_nws1 · 2022-07-11

**Rating:** 4
**Confidence:** 4
**Soundness:** 2 fair
**Presentation:** 2 fair
**Contribution:** 2 fair

**Summary:**

This paper investigated the variance-based regularization method for domain generalization. In particular, the authors originally study how the empirical efficient estimator ($\hat{\mathbf{q}}$) effects the generalization performance for a convex hull of domains. The authors first give an generalization bound for in-distribution generalization based on the in-domain variance and out-of-domain variance (Thm. 1). Then, they study the relationship of empirical efficient estimator and the group DRO objectives, and show the equivalence between group DRO and the variance-based regularization (Thm. 3).

Overall, this paper gives some new insights for the variance-based domain generalization, such as the effects of empirical efficient estimator and the in-distribution generalization bound based on variance. However, I still think the theorical results in this paper is insufficient for answer the question: "Can Variance-Based Regularization Improve Domain Generalization?".  Besides, the main contribution of studying the effect of empirical efficient estimator seems a bit trivial, since there exists work on variance-based domain generalization [1][2].

[1] Krueger et al.. Out-of-distribution generalization via risk extrapolation (rex). ICML 2021.
[2] Xie el tal., Risk variance penalization. arXiv preprint arXiv:2006.07544, 2020.


**Questions:**

I hope the authors to clarify my concerns, but if I am wrong, please correct me.

**Ethics Review Area:**

["I don’t know"]

**Strengths And Weaknesses:**

**Strengths**
1. some new insights for the variance-based domain generalization, such as the effects of empirical efficient estimator and the in-distribution generalization bound based on variance.
2. Although there exists work on variance-based generalization [3]. the study of variance-based domain generalization is interesting to me.
3. The theoretical derivations are complete.

[3] Duchi et al., Variance-based regularization with convex objectives 2017

**Weaknesses**
1. The main concern to me is the significance of the theoretical results. Thm 1 gives an in-domain generalization result (bound the generalization risk of training domains), which is far away from the setting of domain generalization, where the target domain is unseen during training and the discrepancy between the target domain and training domains can be large. Although the authors have discussed this problem in Eq. (9), I still think they should give a generalization bound on the standard setting of domain generalization.

2. Another major concern is the novelty of this paper. To my knowledge, Xie et al. [2] have proposed the relationship between group DRO objective and the variances. I think the novelty of this paper is the study of empirical efficient estimator for variance-based domain generalization, which seems a bit minor.

3. For experimental results in Tab. 1, I notice the results on PACS are amazingly strong (96.9%). The details of experimental setting and analysis for the results is missing, but I think the experiment is important for this work.

---

> ### Author Response · Authors · 2022-08-02
> **Reply to Reviewer nws1 (Part 1)**
>
>
> We thank you for your time and valuable feedback. We answer the questions here.
>
> ‘**W**’ for Weakness, ‘**Q**’ for Question, and '**Ans**' for Answer.
>
> ---
>
> **W1.** The main concern to me is the significance of the theoretical results. Thm 1 gives an in-domain generalization result (bound the generalization risk of training domains), which is far away from the setting of domain generalization, where the target domain is unseen during training and the discrepancy between the target domain and training domains can be large. Although the authors have discussed this problem in Eq. (9), I still think they should give a generalization bound on the standard setting of domain generalization.
>
> **Ans.** Thank you for the question. The domain generalization problem we considered in this work is consistent with the standard setting of domain generalization. We formulate the domain generalization problem into a rigorous and practical form. Furthermore, our theoretical results illustrate the benefits and risks of using variance-based regularization. We answer this question from the following aspects.
>
> - **Setting.** Domain generalization is ultimately a generalization problem from the training data to test data. The concept "domain" is a tool for describing the structure of training data. Since all we see in practice is multi-source training data, we formulate the domain generalization problem as: (1). There exists a domain distribution; (2). The shift in domain distribution causes the distribution shift between training and test data. This formulation is a practical setting for domain generalization with minimal additional assumptions.
>
> - **Discrepancy between Domains.** "The discrepancy between the target domain and training domains can be large" is an ambitious goal of domain generalization. Generally, the discrepancy refers to the distance between the data distributions of two domains. In fact, domain generalization relies on domain similarity. We say the target domain is far from the training domains because the difference between domains is not quantified properly. For example, invariant learning methods aim to discover feature spaces where the domain difference is zero. Here we use the distance between domain distributions to represent the discrepancy between the training and test data. This setting is in line with practical domain generalization tasks.
>
> - **Assumptions.** In general,  achieving the ambitious goal of domain generalization requires strong additional assumptions (or prior information), e.g. certain invariance exists across domains. However, the cost and risk of using these assumptions have not been well studied. In this work, we prove the generalization guarantee of variance-based regularization methods without prior information (Theorem 1) and figure out the risk of using variance-based regularization (Line 200-Line 204).
>
> - **ID Generalization.** We investigate the ID generalization for the following reasons. (1) The ID error is strongly correlated with the OOD error [1].  (2) The ID generalization can reflect the risk of using a domain generalization method. (Please see **Assumptions.**) (3). We consider the standard domain generalization task without additional prior information. Under the standard settings, studying ID error is justified and OOD error relies on guessing the test data (Line 173 - Line 184).
>
> ---
>
> **Reference:**
>
> [1] Miller, et al., Accuracy on the line: on the strong correlation between out-of-distribution and in-distribution generalization. ICML 2021.

---

> > ### Author Response · Authors · 2022-08-02
> > **Reply to Reviewer nws1 (Part 2)**
> >
> > ---
> >
> > **W2.** Another major concern is the novelty of this paper. To my knowledge, Xie et al. [2] have proposed the relationship between group DRO objective and the variances. I think the novelty of this paper is the study of empirical efficient estimator for variance-based domain generalization, which seems a bit minor.
> >
> > **Ans.** This work is different from [2] in the following aspects:
> >
> > - **Data Generation.** The work [2] uses a two-stage data generation. The first stage generates domains, and then the second stage generates the in-domain data points. Here we treat the domain label as an endogenous character and use a one-stage data generation process. Please see Line 21-23.
> >
> > - **Applicability.** Our setting is more suitable for explaining the existing variance-regularized methods. [2] have proposed the relationship between group DRO objective and the standard deviation of risk. But their results fail to explain the utility of penalizing risk variance (REx, [3]). Under our settings, we point out the utility of REx from the view of hypothesis testing (Section 2.2, Line 117 - Line 135).
> >
> > - **Theoretical Results.** [2] focuses on the optimization properties. In this work, we study both generalization and optimization. Our results show the benefits of the weighting correction and the risk of using variance-regularized methods. We also prove the optimization equivalence between group DRO and the proposed method.
> >
> > - **Technical Details.** The technical details of our work are quite different from [2]. In the settings of [2], the two stages of data generation are disentangled. Therefore, the results in [2] (Theorem 3 and Theorem 4 in [2]) are derived by directly combining the inter-domain generalization error and the intra-domain sample-level generalization error. In our work, the data generation is one-stage. So our results (Theorem 1 and Theorem 4) are obtained by decomposing the risk variance (Line 152 - Line 153) and analyzing the generalization error of the pooled training data ($\hat R(f)$ and $R(f)$).
> >
> > ---
> >
> > **W3.** For experimental results in Tab. 1, I notice the results on PACS are amazingly strong (96.9%). The details of experimental setting and analysis for the results is missing, but I think the experiment is important for this work.
> >
> > **Ans.** Tab. 1 reports the ID generalization accuracy to verify the theoretical results in Theorem 1. We can find that the weighting correction improves the ID generalization and the improvement is statistically significant. Please refer to Line 393 - Line 424 for the details of the experimental setting.
> >
> > ---
> >
> > **Reference:**
> >
> > [2] Xie, et al., Risk variance penalization. arXiv preprint arXiv:2006.07544, 2020.
> >
> > [3] Krueger, et al., Out-of-distribution generalization via risk extrapolation (rex). ICML 2021.

---

> > > ### Author Response · Authors · 2022-08-09
> > > **Response to Reviewer nws1**
> > >
> > > Hello Reviewer nws1, we would be grateful if you could confirm whether our response has addressed your concerns and let us know if any issues remain. To recap our response,
> > >
> > > - We formulate domain generalization into a rigorous and practical form. Our theoretical results illustrate the benefits and risks of using variance-based regularization. We explain the significance of our theoretical results in terms of settings, discrepancy between domains, assumptions, and ID generalization.
> > >
> > > - We explain the novelty of our work. We treat the domain as an endogenous character and consider one-stage data generation. Different from Xie et al. (2020), our results cover generalization and optimization, and can explain the utility of rex (Krueger, et al. 2021). The optimization results look similar to those of Xie et al. (2020), but the technical details of the main results are very different.
> > >
> > > - Table 1 validates the results of Theorem 1 and reports the ID accuracy.

---

> ### Comment · Reviewer_nws1 · 2022-08-09
> **Post-Rebuttal**
>
> Thanks for the detailed response!
>
> For W1, I encourage the authors give a generalization for the OOD, because the main contribution of this paper is the OOD generalization.
>
> For W2, I encourage the authors give detailed comparison on the theoretical result to the previous work, e.g., convergence rate, empirical terms, etc.
>
> For W3, I understand that it is the ID setting, which makes me convinced.
>
> Due to the efforts of authors and the detailed response, I would like to raise my score (3->4), but I still think the paper is marginally below the acceptance threshold, I am sorry for that.

---

> > ### Author Response · Authors · 2022-08-09
> > **Response to Reviewer nws1**
> >
> > Thank you for raising the score. In the revision, we will clarify that previous work does not prove generalization guarantees and emphasize the novelty and significance of our theoretical results.

---

### Official Review · Reviewer_18XN · 2022-07-11

**Rating:** 5
**Confidence:** 3
**Soundness:** 3 good
**Presentation:** 3 good
**Contribution:** 4 excellent

**Summary:**

This work studied a variance-based regularization method based on weighting correction for domain generalization. It also theoretically analyzed the potential benefits of the proposed method over ERM, and the connection between the proposed variance-based regularization method and the group DRO problem from an optimization perspective.

**Questions:**

(1) In section 4, how would the selection of the anchor distribution affect the generalization results?
(2) More empirical comparison between the proposed method and ERM can be better for supporting the observations in section 3.1.

**Limitations:**

The authors showed that no potential negative societal impact was observed in this work.

**Strengths And Weaknesses:**

Strengths:
(1) This paper proposed a weighting correction scheme for variance-regularized domain generalization methods inspired by the generalization error bound.
(2) It theoretically compared the proposed method with ERM.
(3) It demonstrated that the objective function of the proposed method is equivalent to a group DRO problem from an optimization perspective.

Weaknesses:
(1) More experiments need to be provided to validate the observations of this paper.
(2) It is unclear how the generalization gap between the training and test data in Line 101-102 is derived.
(3) The computational complexity of variance-regularized domain generalization method with weighting correction scheme is not analyzed.

---

> ### Author Response · Authors · 2022-08-02
> **Reply to Reviewer 18XN**
>
>
> Thank you for your time and insightful comments. We are grateful for your recommendations to improve the manuscript. We answer your questions here.
>
> ‘**W**’ for Weakness, ‘**Q**’ for Question, and '**Ans**' for Answer.
>
> ---
>
> **W1.** More experiments need to be provided to validate the observations of this paper.
>
> **Ans.** Thank you for the comments. Appendix A shows experimental results to validate our theoretical results. Theorem 1 (Line 164) states that the scheme of weighting correction has the ID generalization guarantee. In our experiments, we show that the weighting correction improves the ID prediction accuracy and the improvement is statistically significant (Table 1, Appendix A).
> We will explain the empirical evidence and add citations in the revision.
>
> ---
>
> **W2.** It is unclear how the generalization gap between the training and test data in Line 101-102 is derived.
>
> **Ans.** In the example of Risk Interpolation, the training distribution is $P = \sum_{i=1}^n q_i P_{e_i}$ and the test distribution is $P* = \sum_{i=1}^n q_i* P_{e_i}.$ Then the generalization gap is
>
> $
> err_f = \sum_{i=1}^n q_i* R(f|e_i)  - \sum_{i=1}^n q_i R(f|e_i) = \sum_{i=1}^n ( q_i* -  q_i) R(f|e_i).
> $
>
> In addition, $\sum_{i=1}^n (q^*_i -  q_i)=0$. Then we have
>
> $
> err_f = \sum_{i=1}^n ( q_i* -  q_i) R(f|e_i) = \sum_{i=1}^n ( q_i* -  q_i) \Big( R(f|e_i) - \sum_{i=1}^n q_i R(f|e_i) \Big).
> $
>
> ---
>
> **W3.** The computational complexity of variance-regularized domain generalization method with weighting correction scheme is not analyzed.
>
> **Ans.** Thank you for the comments. The weighting correction scheme does not increase computation cost because we can estimate the empirical domain distribution with the statistical information of the training data. The computational complexity of variance-regularized domain generalization methods is still an open problem because the objective function is non-convex and the optimization procedure is intractable. We will consider this challenging problem in future work.
>
> ---
>
>
> **Q1.** In section 4, how would the selection of the anchor distribution affect the generalization results?
>
> **Ans.** In the general version, the anchor distribution should be determined by prior information. The proper selection of the anchor distribution can improve the OOD generalization, while the wrong selection may cause disastrous results. On the other hand, any anchor distribution, except the empirical training domain distribution, will hurt the ID generalization in terms of asymptotic efficiency.
>
> ---
>
> **Q2.** More empirical comparison between the proposed method and ERM can be better for supporting the observations in section 3.1.
>
> **Ans.** Thank you for the comments. Our theoretical results show that the proposed method can outperform ERM in ID generalization under certain conditions. In Appendix A, we report the empirical results of the proposed weighting correction method (Table 1, Line 426).
>
> Here we provide an example result for better comparison. (Limited by computational resources, we only run one seed per hyperparameter. We will complete the experiments in the revised version.) The experimental results show that our method can outperform ERM in terms of ID generalization accuracy, which is consistent with Corollary 2 in Section 3.1.
>
> ***PACS, ID generalization***
> |method|A|C|P|S|Avg|
> |:---|----|----|----|----|----|
> |ERM|96.5|95.9|96.4|97.2|96.5|
> |Our|96.9|97.0|96.4|97.2|96.9|
>
> ***VLCS, ID generalization***
> |method|C|L|S|V|Avg|
> |:---|----|----|----|----|----|
> |ERM|80.6|87.1|83.9|82.5|83.5|
> |Our|81.9|87.6|85.7|83.4|84.7|
>
>
> ---

---

> > ### Author Response · Authors · 2022-08-09
> > **Response to Reviewer 18XN**
> >
> > Hello Reviewer 18XN, we are thankful for your time and insightful comments. We hope that our responses have addressed your concerns. Please let us know if there remain concerns or if you have more questions. We are more than happy to provide additional clarification.
> >
> > To recap our response,
> >
> > - Appendix A empirically validates Theorem 1. In addition, we report empirical comparisons between the proposed method and ERM.
> >
> > - The generalization gap for the risk interpolation example stems from the difference between the weighted sums of domain risks.
> >
> > - The weighting correction scheme does not increase computation cost.

---

> > > ### Comment · Reviewer_18XN · 2022-08-09
> > > **Thank you for your reply**
> > >
> > > Thank you very much for your reply. The answer to W2 is clear to me now. But for others, I still have the same concerns.
> > >
> > > (1) It is confusing why estimating the empirical domain distribution with the statistical information of the training data will not increase computation cost.
> > >
> > > (2) It seems that anchor distribution (e.g., the estimation quality of empirical training domain distribution) significantly affects the generalization results. This might limit its applications in practice.
> > >
> > > (3) It is hard to conclude from the reported results that the proposed method can outperform ERM in terms of ID generalization accuracy, because they are quite similar results in most cases.

---

> > > > ### Author Response · Authors · 2022-08-10
> > > > **Response to Reviewer 18XN**
> > > >
> > > > Thank you for your further comments. We are sorry that our response did not provide you with a full discussion of the theory and experimental results. We hope our further response can address your concerns.
> > > >
> > > > > It is confusing why estimating the empirical domain distribution with the statistical information of the training data will not increase computation cost.
> > > >
> > > > **Ans.** The empirical domain distribution is estimated by
> > > > $\hat q = (\frac{m_1}{m}, \frac{m_2}{m}, \cdots, \frac{m_n}{m})$
> > > > with $m = \sum_{i=1}^n m_i.$ Here $m$ is the total sample size and $m_i$ is the sample size of the $i$-th domain. The estimation can be done before the training procedure starts. In addition, the computational complexity of estimating $\hat q$ is negligible compared to the training process.
> > > >
> > > > > It seems that anchor distribution (e.g., the estimation quality of empirical training domain distribution) significantly affects the generalization results. This might limit its applications in practice.
> > > >
> > > > **Ans.** Thank you for the comments. The anchor distribution can significantly affect the generalization results. This does not limit the application of variance regularization methods but rather illustrates the importance of our theoretical results. The optimal choice of the anchor distribution depends on known test data. However, access to test data is impossible under a domain generalization setting. Therefore, we infer the choice of the anchor distribution with the training data. Our theoretical results clearly show that without prior information, we should use the empirical domain distribution as the anchor distribution, while applying other domain distributions brings additional risks.
> > > >
> > > > > It is hard to conclude from the reported results that the proposed method can outperform ERM in terms of ID generalization accuracy because they are quite similar results in most cases.
> > > >
> > > > **Ans.** Lines 194-204 discuss when the proposed method is better than ERM and when ERM is better. We study the convergence rate of the excess risk as $m$ tends to infinity. When $m$ is large enough, the method with a faster convergence rate can outperform the other one.
> > > >
> > > > - At Line 194-199, we assume there exists an optimal solution $f* \in \arg\min_f R(f)$ such that $V(f*)=0$. In this case, the convergence rate of the proposed method is faster than that of ERM.
> > > >
> > > > - At Line 200-204, we assume there is no optimal function $f^* \in \arg\min_f R(f)$ satisfies $V(f^*)=0$. In this case, the convergence rate of the proposed method is slower than that of ERM.
> > > >
> > > > These theoretical results fill a void: What are the risks of using variance regularization for domain generalization?

---

### Meta-Review · Area_Chair_HdPa · 2022-08-26

**Recommendation:** Reject
**Confidence:** Certain

**Metareview:**

This paper has been widely discussed between reviewers and authors. Unfortunately, even after the reviewers updated their scores, the paper was still judged to be below the acceptance threshold. I encourage the authors in taking into account the reviewers' comments while preparing the next iteration of their work.

**Award:**

No

---

### Decision · Program_Chairs · 2022-09-14

Reject